# Human Tick-Borne Diseases and Advances in Anti-Tick Vaccine Approaches: A Comprehensive Review

**DOI:** 10.3390/vaccines12020141

**Published:** 2024-01-29

**Authors:** Marie-Edith Nepveu-Traversy, Hugues Fausther-Bovendo, George (Giorgi) Babuadze

**Affiliations:** 1Global Urgent and Advanced Research and Development, Batiscan, QC G0X 1A0, Canada; mntraversy@guardrx.org; 2Department of Microbiology & Immunology, University of Texas Medical Branch, Galveston, TX 75550, USA; hffausth@utmb.edu

**Keywords:** tick-borne pathogen, emerging viruses, tick-borne disease, emerging infectious disease, vaccine development, anti-tick vaccine

## Abstract

This comprehensive review explores the field of anti-tick vaccines, addressing their significance in combating tick-borne diseases of public health concern. The main objectives are to provide a brief epidemiology of diseases affecting humans and a thorough understanding of tick biology, traditional tick control methods, the development and mechanisms of anti-tick vaccines, their efficacy in field applications, associated challenges, and future prospects. Tick-borne diseases (TBDs) pose a significant and escalating threat to global health and the livestock industries due to the widespread distribution of ticks and the multitude of pathogens they transmit. Traditional tick control methods, such as acaricides and repellents, have limitations, including environmental concerns and the emergence of tick resistance. Anti-tick vaccines offer a promising alternative by targeting specific tick proteins crucial for feeding and pathogen transmission. Developing vaccines with antigens based on these essential proteins is likely to disrupt these processes. Indeed, anti-tick vaccines have shown efficacy in laboratory and field trials successfully implemented in livestock, reducing the prevalence of TBDs. However, some challenges still remain, including vaccine efficacy on different hosts, polymorphisms in ticks of the same species, and the economic considerations of adopting large-scale vaccine strategies. Emerging technologies and approaches hold promise for improving anti-tick vaccine development and expanding their impact on public health and agriculture.

## 1. Introduction

Ticks are arthropods and belong to the class Arachnida, distinguishing themselves from insects. Specifically, they belong to the order Ixodida and are obligate hematophagous ectoparasites that feed on the blood of one or more hosts, including mammals, birds, reptiles, and humans. They depend entirely on their hosts to complete their life cycle. Following mosquitoes, they rank second as vectors of human diseases. Their role in transmitting animal diseases is well established [1]. Ticks notably surpass all other arthropods in their capability to transmit a diverse array of infectious agents, including viruses, bacteria, and parasites. Among these, tick-borne viral diseases have become a growing concern in both medical and veterinary fields. This concern arises from the expanding geographic range of tick species, the emergence of outbreaks in previously unaffected regions due to migratory birds, shifts in global socioeconomic and climatic factors, and the absence of effective control measures [2,3].

While the primary approach for tick control has traditionally relied on the application of chemicals known as acaricides, there is a growing concern over the rising instances of tick resistance to these chemicals. Furthermore, there are apprehensions regarding the accumulation of acaricide residues in meat, milk, and the environment [4,5]. In the context of a comprehensive tick management strategy, the development of anti-tick vaccines targeting tick-borne pathogen reservoirs is acknowledged as a promising approach. These vaccines aim to block the transmission of pathogens to other hosts, including humans, thereby enhancing control measures against ticks [6].

The objective of this review is to offer a concise summary of the current understanding of established and emerging tick-borne pathogens, emphasizing the global impact of these pathogens on human health. Specifically, it aims to underscore the significance of anti-tick vaccines as a promising preventive strategy designed to interrupt the enzootic cycle of these pathogens, thereby mitigating the risk of transmitting multiple diseases.

## 2. Tick Overview

There are an estimated up to 900 tick species known to exist. All ticks are classified into three main families: Argasidae (186 species), Ixodidae (692 species), and Nuttalliellidae, which consists of a single species [7,8]. The complete taxonomy classification is depicted in Figure 1. The most significant family, Ixodidae (hard ticks), is characterized by a tough, sclerotized exoskeleton, with a dorsum that is partially or entirely covered with chitin. They possess a scutum, a hardened shield-like structure, located in the anterior part of their body [9]. Hard ticks typically attach to their hosts for several days during feeding. The Argasidae (soft ticks) have a dorsum lacking chitin and possess a leathery, flexible exoskeleton [10]. Unlike hard ticks, they lack the scutum. Soft ticks typically feed for a shorter duration compared to hard ticks [11]. As for the Nuttalliellidae family, it is relatively obscure and consists of a single representative, *Nuttalliella namaqua*.

Ticks can also be classified based on their host preferences. Ixodid ticks can exhibit a feeding pattern often referred to as three-host ticks, involving three hosts during their life cycle stages (larva, nymph, and adult). However, it is important to note that not all ixodid ticks strictly follow this pattern. On the other hand, argasid ticks often exhibit multi-host feeding behavior, acquiring multiple blood meals during their development [8]. Additionally, based on a morphological characteristic known as the hypostome, a structure in the mouthparts used for feeding, the Ixodidae family can be further classified into two groups: the Prostriata group (possessing a hypostome), which includes the genus *Ixodes*; and the Metastriata group (lacking a hypostome), encompassing all other genera within the Ixodidae family (Figure 1).

Ticks serve as highly effective vectors for numerous pathogens due to their ability to interact with various vertebrate hosts throughout their life cycle. Hard ticks undergo a life cycle characterized by three active feeding stages: larvae, nymphs, and adults. Typically, the larval and nymphal stages of these ticks feed on small mammals, especially rodents, which serve as ideal animal reservoirs for numerous tick-borne pathogens. Moreover, several tick species are known to feed on ground-feeding birds, contributing to the dissemination of pathogens that affect both humans and animals [12,13,14]. Hard ticks have a cosmopolitan distribution and exhibit diverse host-seeking behaviors; they typically engage in prolonged feeding periods, which can vary from 3 to 12 days. The specific duration of feeding depends on the tick species and its developmental stage [9]. Unlike hard ticks, soft ticks have rapid feeding habits, typically completing their blood meals in a matter of minutes to an hour. Additionally, they have the capacity to take multiple blood meals during each developmental stage [15]. Soft ticks transmit fewer human pathogens than hard ticks due to their feeding behavior and limited habitat [16]. The likelihood of disease transmission correlates with the duration of tick attachment. Additionally, the nature of the pathogen plays an important role in the speed of transmission between ticks and hosts [17].

It is important to highlight that in the transmission of human pathogens, the nymph and adult stages of ticks play a pivotal role. Meanwhile, larvae, which are six leg ticks, primarily contribute to transovarial transmission—the transmission from parent to offspring via the ovaries. However, it is important to note that not all pathogens are transmitted in this manner. Larvae can also acquire pathogens while feeding on hosts and subsequently transmit it to the next host, as previously observed with spirochetes involved in Lyme disease [18]. Their small size makes them challenging to detect, amplifying their potential threat.

While some ticks are host specific, others can exhibit a two-host or three-host feeding patterns. For example, *Rhipicephalus microplus* is a one-host tick; all developmental stages feed and mature on the same host, especially on cattle. However, many ticks exhibit opportunistic and generalist feeding behaviors. *Amblyomma americanum*, for instance, feeds on mammals, birds, and reptiles, similar to *Ixodes ricinus*, which is also known for its extensive host range.

In the case of three-host ticks, the process unfolds as follows: First, the larvae feed on a small host, typically rodents. Then they detach, undergo molting to become nymphs. These nymphs subsequently attach to a different host, feed once again, detach, and ultimately, the adult tick attaches to a third, larger host to complete its feeding cycle. Ticks find their hosts by detecting the breath and body odors of an animal, sensing body heat, moisture, or vibrations. The time required for a tick attachment varies depending on its species and life stage, ranging from less than 1 h to 7 days [19]. Once it identifies a suitable feeding location, the tick secures itself to the skin and makes an incision into the surface. While feeding, ticks release substances that serve various purposes, including anchoring themselves to the host, acting as an anesthetic to conceal the pain of their bite, and inhibiting the coagulation of blood. This substance is termed “cement” [20]. Following the bite of an infected tick and a brief incubation period, the initial stage of infection typically manifests as a non-specific febrile illness, characterized by a variety of symptoms [21]. As obligatory parasites, ticks require feeding on animals, including important reservoirs of pathogens like rodents, to reproduce and survive. Due to their efficient feeding and persistent nature, once attached, ticks pose a substantial risk for disease transmission to larger hosts, such as humans. 

Furthermore, tick-borne diseases pose a significant threat to livestock, impacting the health and productivity of farming animals worldwide. These diseases, transmitted by ticks, affect a wide range of animals, including cattle, sheep, goats, and horses, leading to serious health issues and economic losses. Among the most prevalent tick-borne diseases in livestock are anaplasmosis, babesiosis, theileriosis and Lyme borreliosis. Of note, tick-borne diseases of veterinary relevance have been reviewed [22]. The economic impact of these diseases is substantial, as they can lead to decreased meat and milk yields, increased veterinary costs, and in severe cases, animal fatalities. Effective control of tick populations is crucial in mitigating the impact of these diseases [23]. This includes the use of acaricides, tick-resistant breeds, pasture management strategies, and the use of the only commercially available anti-tick vaccine, namely, Gavac^®^.

The transition from discussing animal diseases to those affecting humans reflects the broader context of tick-borne diseases, which span multiple species and ecosystems. However, although TBDs are of veterinary relevance, this review aims to provide an overview of the extent of the threat associated with human tick-borne diseases. It will primarily focus on tick-borne pathogens from each microorganism group (viruses, bacteria, and parasites) that have the potential to spread over extensive geographical areas and pose a significant health burden.

## 3. Tick-Borne Viral Infections

Ticks transmit a variety of viruses that pose significant concerns for both public and veterinary health. The expansion of tick populations, along with a rising incidence of tick-borne diseases, has garnered attention in the field of global public health. The intricate interplay between tick-borne viruses (TBVs) and ticks has resulted in a complex relationship wherein the virus life cycle is perfectly coordinated with the tick feeding cycle. This synchronization enables ticks to harbor the virus for extended periods without any noticeable effect on their biological processes [24]. Some TBVs have the potential to induce severe illnesses in both humans and animals, including central nervous system (CNS) conditions like meningitis, meningoencephalitis, or encephalomyelitis, as well as hemorrhagic diseases. Meanwhile, some TBVs lead to less severe outcomes or are sporadically documented. They exhibit remarkable taxonomic diversity, encompassing five significant viral families circulating between ticks and vertebrate hosts: *Flaviviridae*, *Nairoviridae*, *Phenuiviridae*, *Orthomyxoviridae*, and *Reoviridae* (Table 1). The geographical distribution of TBVs and associated tick vectors is indicated in Figure 2.

### 3.1. Flaviviridae

The genus *Flavivirus*, termed *Orthoflavivirus* since 2023 [32], belongs to this family and constitutes a significant subgroup of tick-borne viruses. These are small, single-stranded RNA viruses characterized by a positive-sense genome enclosed in a lipid envelope derived from the host [33]. This subgroup includes viruses such as tick-borne encephalitis virus (TBEV), Powassan virus (POWV), Asian Omsk hemorrhagic fever virus (OHFV), Kyasanur Forest disease virus (KFDV), Alkhurma virus (ALKV), and Louping Ill virus (LIV) (which causes rare human cases). These viruses are notable for their ability to trigger encephalitis and hemorrhagic symptoms. They are prevalent in Europe, Russia, Japan, and China, with more than 10,000 reported cases annually. The diseases induced by the orthoflaviviruses encompass a spectrum, ranging from asymptomatic or mild febrile illness to severe conditions like hemorrhagic fever or profound encephalitis, often accompanied by substantial morbidity and mortality [20,34].

Over the past 40 years, there has been a noticeable increase in the incidence rate of tick-borne encephalitis, accompanied by an expansion in the endemic range of the virus [35,36]. Tick-borne encephalitis is attributed to three main genetically distinct subtypes of viruses within a single TBE virus (TBEV) serocomplex. These three subtypes are the Siberian subtype (TBEV-Sib), the Far Eastern subtype TBEV (TBEV-FE), and the European subtype (TBEV-Eu) [37]. Illness caused by the TBEV-FE is generally more severe than that caused by TBEV-Eu, with the latter having the majority of infections appearing to be subclinical, while a minority can progress to a neurological phase. Additionally, the case fatality rate associated with TBEV-FE can be up to 30%, whereas for TBEV-Eu, it is approximately 2%. On the other hand, the Siberian subtype TBEV-Sib exhibits intermediate disease severity but has been connected to chronic infections [38]. The main tick vectors for TBEV-Eu are *Ixodes ricinus*, while *I. persulcatus* serves as the vector for TBEV-Sib and TBEV-FE [39]. Other tick genera including *Haemaphysalis*, *Dermacentor*, *Hyalomma*, and *Rhipicephalus* are acknowledged vectors in Asia as they can carry and transmit the virus. The main reservoir hosts are rodents and deer [40].

Powassan virus (POWV) is the only member known to have an established endemic presence in North America. Initially isolated in northeast Canada in 1958, it has since been identified in northern regions of the United States, southern Canada, and even in far-eastern Asia [41,42]. POWV disease is asymptomatic in most people but can cause devastating and fatal encephalitis. *Ixodes scapularis* and *I. cookei* ticks are the primary vectors for the POWV transmission in America, while *Haemaphysalis longicornis* is known to transmit POWV in Siberia [43]. The main reservoir hosts are small mammals, especially rodents [44].

OHFV is currently only known in western Siberia, Russia. The initial symptoms include non-specific flu-like symptoms such as headache, cough, nausea, muscle pain, and chills. A second phase can occur in some patients, leading to encephalitic symptoms such as headaches and meningitis. OHFV is transmitted to humans by tick bites, mainly by *Dermacentor reticulatus* and *D. marginatus*. The main reservoir hosts also include small rodents, such as muskrats [45].

KFDV and AHFV cause life-threatening hemorrhagic fever, with fatality rates of up to 20% for AHFV. KFDV was discovered in India, while AHFV was found in Saudi Arabia and Egypt. The main tick vectors involved in the transmission of KFDV and AHFV are *Haemaphysalis spinigera* and *Ornithodoros savignyi* (soft tick), respectively. KFDV has been isolated from rodents, birds, cattle, and bats, while AHFV has been linked with camels and sheep [46].

LIV is an acute viral and deadly disease that mostly affects livestock but can affect humans. It is transmitted by *Ixodes ricinus*, the same vector as TBEV. However, in contrast to TBEV, whose main hosts are rodents, the main reservoir hosts for LIV are sheep and red grouse [47].

Most orthoflavivirus infections are not totally preventable by vaccines and cannot be treated with therapeutic drugs. However, the Ticovac (Baxter Hyland Immuno, Vienna, Austria) vaccine against TBEV was approved in 2000 in Europe and by the FDA in 2021 [48]. It also confers some immunity to OHFV and LIV [49]. There is also a vaccine with limited efficacy for KFDV [46]. In general, the recommended approach to treatment primarily involves providing supportive care.

### 3.2. Nairoviridae

The Crimean–Congo hemorrhagic fever virus (CCHFV) is an emerging tick-borne virus belonging to the Bunyavirales order and *Nairoviridae* family. The viral genome of CCHFV is tri-segmented and consists of single-stranded negative-sense RNA [50]. It is widely spread around the world and has a wide range of hosts and tick vectors in Eurasia and Africa, with *Hyalomma marginatum* being the most proficient vector among 15 proven vector species. CCHF typically starts with non-specific symptoms but can rapidly progress to a severe, and in some cases, fatal disease. The transmission results in a disease characterized by symptoms such as fever, headache, myalgia, and hemorrhagic manifestations [51]. As the initial symptoms are non-specific and infections often go unnoticed, patients frequently seek medical attention when the disease has already reached advanced stages.

This virus is known to have endemic regions in Africa, Asia, the Middle East, and southeastern Europe. The transmission and life cycle of CCHFV have been thoroughly documented; however, it continues to be a significant emerging pathogen across a substantial portion of its geographical range. Transmission involves various modes, including tick-borne transmission and nosocomial and direct transmission from infected livestock. Livestock play a definitive role in maintaining CCHFV transmission, with cattle and sheep being primary candidates for reservoir hosts, along with goat, hares, and many other mammals [52]. It is noteworthy to highlight that in the past two decades, CCHFV has expanded its presence to areas and nations that were previously untouched, with Spain being one such example [50]. Until now, it has never been reported in northern Europe, Australia, or in the Americas.

The elevated mortality rate and the lack of effective drugs or licensed vaccines compound the concerns surrounding CCHF. Because of these factors, CCHF is included on the priority list of the WHO for Research and Development, as well as on the US National Institute of Allergy and Infectious Diseases (NIH/NIAID) priority A list, signifying a disease with the highest level of risk to both national security and public health [53,54].

### 3.3. Phenuiviridae

Other TBVs members of the Bunyavirales order belong to the *Phenuiviridae* family, more specifically tick-borne bandaviruses. Their genome structure possesses a segmented, negative-strand RNA genome composed of large (L), medium (M), and small (S) segments [55,56,57]. Three distinct members can be highlighted: severe fever with thrombocytopenia syndrome virus (SFTSV), Heartland virus (HRTV), and Bhanja viruses (BHAV). One of the recently emerging tick-borne infectious diseases is caused by SFTSV. This novel bandavirus was initially reported in China in 2009; however, cases of SFTS were reported in Japan and Korea in 2012, and more recently, cases have emerged in Vietnam and Taiwan. It has been officially named Dabie bandavirus and is classified within the genus *Bandavirus* [58]. The specific lifecycle and mechanisms responsible for the sustained transmission of SFTSV in nature are still not fully understood, but they involve vertebrate hosts. Potential reservoirs include a variety of both domestic and wild animals [59]. Moreover, it is widely recognized that transmission through Asian tick species, particularly *Haemaphysalis longicornis*, *Amblyomma testudinarium*, *Dermacentor nuttalli*, *Hyalomma asiaticum*, and *Ixodes nipponensis*, are considered the main vectors responsible for transmitting SFTSV [60,61,62,63]. Clinical symptoms of SFTSV infection primarily manifest as high fever and thrombocytopenia. Additional clinical manifestations may include gastrointestinal disorders, leukocytopenia, and a tendency towards hemorrhagic symptoms [64]. Currently, there is no standardized therapeutic protocol in place for treating SFTSV infection in patients, and commercial vaccines for this virus have not yet been developed or made available.

In the realm of emerging tick-borne pathogens, the Heartland virus (HRTV), recently reclassified into the *Bandavirus* category, has gained attention. Distinct from its counterparts, HRTV is notorious for inducing severe health complications, primarily characterized by a marked reduction in platelets and lymphocytes, presenting clinical parallels with SFTSV. Its transmission is primarily linked to the lone star tick, *Amblyomma americanum*, a species infamous for its role in spreading diverse pathogens affecting both human and animal health [65]. While the exact natural hosts of HRTV are yet to be determined, animals such as deer, coyotes, and raccoons are currently under suspicion [66]. Reports from the United States indicate an alarming rate of HRTV infections, some leading to critical health states and fatalities. This trend has prompted the National Institute of Allergy and Infectious Diseases (NIAID) to categorize HRTV as a Category C Priority Pathogen, reflecting its significant impact on public health and the urgent need for further research and surveillance.

Unlike SFTSV and HRTV, the Bhanja viruses (BHAV) seem to be categorized into few lineages, which includes African and Eurasian lineages; this distinction may be influenced by, or associated with, the geographic distribution of their tick vectors and potential mammalian reservoirs. Research has indicated the presence of BHAV in multiple species of hard ticks. In terms of geographic distribution and host specificity, there is a noticeable pattern: Eurasian Bhanja group viruses have been predominantly identified in *Haemaphysalis* spp. ticks, whereas their African lineage counterparts have been detected in a variety of tick genera, including *Amblyomma*, *Dermacentor*, *Rhipicephalus*, and *Hyalomma* [67]. As of now, the medical community faces a significant challenge in combating these infections, as there are no established specific treatments or preventive strategies for illnesses caused by bandaviruses. This underscores the urgency for continued research and development in this area.

### 3.4. Orthomixoviridae

The genus *Thogotovirus* within the family *Orthomyxoviridae* comprises multiple species, with only three of them known to be associated with human diseases. These species include Thogoto virus (THOV), Dhori virus (DHOV), and Bourbon virus (BRBV). These viruses are transmitted by various hard tick species [68], and their genome consists of six segments of negative-sense, single-stranded RNA [69]. Thogotoviruses, transmitted by ticks, have recently gained attention due to their significant zoonotic potential. This was notably highlighted by the emergence of BRBV in the United States since 2014 [70]. While cases of human fatalities due to *Thogotovirus* infections are rare, they underscore the potential health risks these viruses pose. The full extent of their zoonotic impact, however, remains largely uncharted. Thogotoviruses are distinguished by their transmission through ticks and their ability to infect a wide range of mammalian hosts, including rodents, large game animals, livestock like sheep and cattle, and even camels [71]. The severity of the illnesses they cause can be highly variable. A notable example is the BRBV cases in the United States, where two individuals in Kansas and Missouri succumbed to the virus in 2015 and 2017, respectively. Their illnesses were marked by severe respiratory distress and liver damage [70,72]. Epidemiological studies conducted by different scientific groups in various countries have shown that THOV can be transmitted by ixodid ticks, and it has been isolated from multiple tick species in different countries. These include *Rhipicephalus* spp. In Kenya; *Amblyomma variegatum*, *Rhipicephalus annulatus*, and *Hyalomma nitidum* in the Central African Republic; *Rhipicephalus bursa* ticks in Italy; *Haemaphysalis longicornis* in Japan; and *Amblyomma americanum*, which is also the vector of BRBV to humans in the USA [73,74,75,76,77,78]. DHOV has also been isolated from various tick species, such as *Hyalomma marginatum* ticks on livestock in Portugal and Russia, as well as from *Rhipicephalus pulchellus* and *Amblyomma gemma* ticks in Kenya [79,80,81,82]. At present, there are no established treatments or preventive measures specifically designed for infections caused by Thogotoviruses.

### 3.5. Reoviridae

Coltiviruses are tick-borne viruses belonging to the genus *Coltivirus* of the family *Reoviridae*. The genus *Coltivirus* includes two members: the New World Colorado tick fever virus (CTFV), which is widespread in the Rocky Mountain region of North America, and Old World Eyach virus (EYAV), first isolated in Europe in 1976 [83]. Their genome consists of 12 double-stranded RNA (dsRNA) segments. Coltiviruses have been isolated from various mammalian species, including humans, as well as from ticks and mosquitoes, which serve as arthropod vectors. However, ticks, particularly those of the species *Dermacentor andersoni*, *D. occidentales*, *D. albipictus*, *D. parumapertus*, *Haemaphysalis leporispalustris*, *Otobius lagophilus* (soft tick), *Ixodes sculptus*, *I. spinipalpis*, *I. ricinus*, and *I. ventalloi*, are the principal vectors for these viruses [84].

CTFV exhibits a broad host range, which encompasses various animals such as ground squirrels, chipmunks, wild mice, wood rats, wild rabbits, hares, porcupines, marmots, deer, elk, sheep, and coyotes [85]. Certain studies, which relied on serologic surveys, have revealed the presence of antibodies to EYAV (Eyach virus) in a wide range of animals. These animals include European rabbits (*Oryctolagus cuniculus*), mice, mountain goats, domestic goats, sheep, and deer [86]. Individuals infected with CTFV typically encounter a rapid onset of several symptoms. These include elevated body temperature, intense headaches, pain in the region behind the eyes, sensitivity to light, muscular soreness, stomachache, and an overall feeling of illness. While most cases are manageable, there are occasional reports of more severe forms of the disease, particularly in young patients. These severe cases may involve complications like infections affecting the central nervous system, hemorrhagic fever, or inflammation of the heart and testes [85]. In a separate but related finding, antibodies targeting the Eyach virus (EYAV), another member of the *Coltivirus* genus, have been identified in patients with certain neurological disorders. The prevalence and ecological dynamics of EYAV, especially in European regions, are still under investigation [87]. Currently, the medical community lacks specific treatments or preventive vaccines for infections caused by *Coltivirus* members.

## 4. Tick-Borne Bacterial Infections

Tick-borne bacterial infections can have significant health impacts due to their ability to cause a range of diseases in humans. These infections are transmitted to humans through the bite of infected ticks, primarily hard ticks, but soft ticks can also be responsible. The importance of understanding and addressing tick-borne bacterial infections lies in their potential to cause significant morbidity and, in some cases, mortality if not diagnosed and treated promptly. These infections can have a profound impact on health, leading to long-term complications and affecting various body systems, including the central nervous system. Most bacterial species (~90%) transmitted by ticks can be categorized into two orders: Spirochaetales and Rickettsiales (Table 1) [12]. The geographical distribution of bacterial pathogens and their related vectors is indicated in Figure 2.

### 4.1. Spirochaetales

Spirochetes involved in tick-borne infections belong to the *Borrelia* genus and are responsible for one of the most concerning tick-borne diseases for public health, namely. Lyme disease, affecting more than half a million people each year. *Borrelia* is also associated with tick-borne relapsing fever and is suspected to be involved in a new syndrome named southern tick-associated rash illness (STARI). *Borrelia* spp. Are Gram-negative spirochetes measuring 20 to 30 µm long, and they possess a linear genome, an exceptional feature for a prokaryote [88].

#### 4.1.1. Lyme Disease

Lyme borreliosis, commonly known as Lyme disease, stands as the most common tick-borne infection, estimated at over 476,000 annual cases in the USA and more than 100,000 cases in Europe [89,90]. Lyme disease is caused by various genetic species (genospecies) within the *Borrelia burgdorferi* sensu lato complex [91,92]. It is transmitted by hard ticks (Ixodidae) of the genus *Ixodes*, including *I. Ricinus* in Europe, *I. persulcatus* in Russia and Asia, and *I. scapularis* or *I. pacificus* in North America [20,93]. The most relevant pathogenic genospecies encompass *B. burgdorferi* sensu stricto, *B. garinii*, *B. afzelii*, *B. bavariensis*, and the recently isolated *B. mayonii*. The last has been detected in patients, adult *I. scapularis* ticks, and rodents in the upper midwest of the USA [94,95,96]. In North America, *the Borrelia* species responsible for Lyme disease are *B. burgdorferi* sensu stricto and *B. mayonii*. Lyme borreliosis is the most prevalent tick-borne infectious disease in North America and moderate-climate regions of Eurasia, signifying its importance in public health within these areas.

Small mammals like mice and voles, along with certain bird species, serve as the primary vertebrate reservoirs for Lyme *Borrelia* spp. In tick environments, deer play a crucial role in sustaining tick populations, as they are one of the few wild hosts capable of feeding enough adult ticks. However, they are not suitable reservoirs for Spirochaetales. Cattle do not support the bacteria, and while sheep are not considered significant reservoir hosts, further research is needed due to the limited available data on this topic [97,98,99]. A relevant factor in identifying reservoir competence is how susceptible the specific genospecies of Lyme *Borrelia* are to being killed by the complement system of the animal host [100].

Lyme disease’s clinical symptoms vary by stage and duration, with Erythema migrans (EM) being the most frequent feature, occurring in over 80% of cases in North America and Europe [101]. Appearing 3 to 14 days following a tick bite, EM is a skin-based reaction to spirochete proliferation, often coupled with fatigue, fever, headache, mild neck stiffness, and joint or muscle pain. Without prompt treatment, the infection can lead to neurological complications (such as facial palsy, meningitis, radiculopathy), cardiac issues (like carditis with atrioventricular block), and arthritis —mainly monoarticular or oligoarticular, typically affecting fewer than five joints [102]. Lyme arthritis, resulting from *B. burgdorferi* sensu stricto infection, is more common in North America, with approximately 60% of untreated patients with EM experiencing this manifestation of the disease. In contrast, only 3–15% of patients affected by Lyme disease in Europe experience Lyme arthritis, where *B. garinii* and *B. afzelii* are more frequently isolated. The neurological manifestation is called Lyme neuroborreliosis and highly correlates with an infection from *B. garinii*, although *B. Burgdorferi* and *B. afzelii* can also be associated with neurological symptoms, albeit to a lesser extent [103]. Concerning the newly identified Lyme disease agent *B. mayonii*, it leads to symptoms such as fever, headache, rash, and neck pain shortly after infection and can result in arthritis within a few weeks. In contrast to *B. burgdorferi*, it can also induce nausea and vomiting, and infected patients exhibit higher spirochetes loads [104].

Conventionally, the primary treatment for Lyme disease involves brief antibiotic regimens. This method has generally been successful in eliminating the infection and enhancing patient well-being. However, recent findings suggest that the benefits of antibiotics may be short-lived, with a notable proportion of patients experiencing a resurgence of symptoms post-treatment. This persistent form of the ailment is often termed ‘chronic Lyme disease’ [105,106,107,108]. The exploration of antibiotic therapies for Lyme borreliosis has not been extensive. In both Europe and the U.S., a handful of randomized studies have assessed various antibiotic strategies targeting specific Lyme disease manifestations such as EM, neuroborreliosis, and Lyme arthritis, as detailed in the IDSA guidelines [109,110]. Approaches to treat less common manifestations typically rely on accumulated case studies or the consensus of specialists in the field. Intriguingly, recent investigations have highlighted the potential of disulfiram monotherapy as a treatment option for Lyme disease patients [111]. In the realm of prevention, vaccines for Lyme borreliosis are in use for animals, but as of now, a human vaccine is not available. A notable attempt was the introduction of a recombinant OspA-based vaccine in the U.S., which was available from 1998 to 2002 [112].

However, concerns arose regarding its safety, particularly its potential relationship to autoimmune arthritis, which led to its eventual withdrawal from the market [113]. A vaccine candidate that combines the six most common OspA serotypes (VLA15, Valneva, France) is currently in a phase 3 clinical trial and shown promising results [114,115].

#### 4.1.2. Tick-Borne Relapsing Fevers

*Borrelia* species are also involved in another disease known as tick borne relapsing fever (TBRF). There is the classical relapsing fever, which involves various *Borrelia* species and is transmitted by soft ticks, mainly *Ornithodoros* and *Carios* species. In addition, there is a relatively new type of relapsing fever caused by another *Borrelia* species transmitted by hard ticks. First described in Japan in 1994, it is named *B.miyamotoi*. It co-circulates with *B. burgdorferi* sensu lato and is transmitted by *I. scapularis* and *I. pacificus* in North America, and *I. ricinus* and *I. persulcatus* in Europe and Asia, respectively, though to a lesser extent [116]. Both relapsing fevers share similar symptoms such as fever, headaches, chills, myalgia, arthralgia, and nausea. However, unlike the classical relapsing fever, *B. miyamotoi* infection does not typically cause symptoms like epistaxis, abortion, jaundice, or major organ failure. Reservoirs hosts are diverse and include small vertebrates such as mice, voles, chipmunks, and certain bird species. Treatments involve antibiotics and follow guidelines used for Lyme borreliosis [117,118].

#### 4.1.3. Southern Tick-Associated Rash Illness (STARI)

This emerging zoonotic disease is transmitted by the lone star tick (*Amblyomma americanum*) in the southeastern and southcentral USA. Similar to the erythema migrans seen in Lyme disease, it causes an annular rash with central clearing. However, the causative agent remains unknown, but it is suspected to be associated with *B. lonestari* [119].

### 4.2. Rickettsiales

Bacteria belonging to the order Rickettsiales are small obligate intracellular parasites of eukaryotes. Many genera of this order are significant emerging and re-emerging pathogens associated with tick-borne infections worldwide. Members belonging to the *Rickettsiacae* family are responsible for rickettsiosis (spotted fevers), while members of the *Anaplasmatacae* cause anaplasmosis, ehrlichiosis, and neoehrlichiosis [120].

#### 4.2.1. Rickettssiacae

Rickettsioses are among the oldest known infectious diseases caused by *Rickettsia* species, which constitute a diverse group present in many vectors including ticks, lice, fleas, mites, and mosquitoes [121]. These *Rickettsia* species are classically divided into the spotted the fever group and the typhus group, with the latter usually more associated with mites, fleas, and lice. These zoonotic pathogens are responsible for infections that spread through the bloodstream to various organs in the body. They can cause mild to serious life-threatening infections, characterized by non-specific clinical symptoms such as fever, headache, myalgias, and rash. Ticks and small rodents are amplifying reservoir hosts [122].

Rocky Mountain spotted fever (RMSF) stands out as the most clinically severe rickettsiosis transmitted by several species of ticks, including the American dog tick (*Dermacentor variabilis*), the Rocky Mountain wood tick (*Dermacentor andersoni*), and the brown dog tick (*Rhipicephalus sanguineus*) [123]. For almost ninety years, *Rickettsia rickettsi* stood out as the most extensively studied and lethal rickettsial species within the spotted fever group, being the only tick-borne *Rickettsia* species linked to human disease in the New World. Other rickettsial species in this region were considered non-pathogenic. Similar patterns, characterized by one known pathogenic rickettsial species coexisting with various species of unknown pathogenicity, have also been observed in Europe and Africa (*Rickettsia conorii*), Asia (*Rickettsia sibirica*), and Australia (*Rickettsia australis*) [123,124]. Over the last three decades, advancements in molecular techniques have led to a substantial rise in identified species within this bacterial group. The ongoing discovery of numerous rickettsial species in diverse invertebrates has raised new questions about their biology, ecology, epidemiology, geographical distribution, and potential pathogenicity [125,126]. Currently, there are over 48 confirmed species of Rickettsia in the spotted fever group, each associated with distinct human infections and geographical regions [121]. Some of these species are listed in Table 1 along with their tick vector and associated disease [127].

The treatment for rickettsiosis typically Involves a range of antibiotics. Historically, tetracyclines or chloramphenicol have been used, although erythromycin and fluoroquinolones are less effective. The preferred and most effective therapy is doxycycline, with chloramphenicol as an alternative option. Several studies have demonstrated that doxycycline shortens the course of Mediterranean spotted fever and leads to a rapid remission of symptoms [128]. While antibiotics can be used to treat rickettsioses, there is a strong desire for a vaccine against *Rickettsia* species for several reasons. Rickettsial diseases are widespread, particularly in impoverished nations, and there are concerns about the emergence of antibiotic resistance. Several attempts were made to create live attenuated and subunit vaccines against rickettsiosis in the 1920s, 1970s, and 1980s. However, these vaccines did not prevent infection, although they significantly reduced the case fatality rate in vaccinated volunteers [129]. Presently, there is no widely available licensed vaccine specifically designed to prevent rickettsial diseases. Developing vaccines for rickettsial diseases such as Rocky Mountain spotted fever has proven challenging due to the complexity of the bacteria and the absence of a universal vaccine capable of covering all strains.

#### 4.2.2. Anaplasmatacae

This family encompass several genera, including *Anaplasma*, *Ehrlichia*, and *Neoehrlichia*, containing species involved in tick-borne diseases. These small, Gram-negative bacteria (ranging from 0.2 to 1.5 µm) are obligate intracellular microbes which amplify within membrane-bound vacuoles [130].

*Anaplasma phagocytophilum*, the causative agent of granulocytic anaplasmosis (HGA), is considered one of the most important species due to its zoonotic potential [131]. *A. phagocytophilum* is responsible for tick-borne fever (TBF) in ruminants and granulocytic anaplasmosis in equines (EGA), canines (CGA), and humans (HGA). Since 1990, the incidence of *A. phagocytophilum* infections has significantly increased in the US. Meanwhile, the geographical distribution of the pathogen and its primary vector, *Ixodes ricinus*, has been expanding its geographical range in terms of latitude and altitude, now covering almost the entire territory of continental and Atlantic Europe [132]. The ecology of *A. phagocytophilum* is becoming better understood. The bacterium is maintained in a transmission cycle involving *I. scapularis* in the eastern United States, *I. pacificus* in the western United States, *I. ricinus* in Europe, and possibly *I. persulcatus* in parts of Asia. Tick infection occurs after an infectious blood meal, and the bacterium is transmitted trans-stadially but not trans-ovarially [133]. *A. phagocytophilum* primary mammalian reservoir in the eastern part of North America is the white-footed mouse, *Peromyscus leucopus*. However, other small mammals and white-tailed deer (*Odocoileus virginianus*) can also carry the bacterium. Human infections happen when people enter habitats shared by ticks and small mammals [134,135]. HGA exhibits clinical variability that spans from asymptomatic infection to fatal disease, but the majority of patients experience a moderately severe febrile illness accompanied by symptoms such as headache, myalgia, and malaise. Extensive studies across North America and Europe, involving up to 685 patients, have revealed common manifestations including malaise, fever, myalgia, thrombocytopenia, leukopenia, anemia, elevated hepatic transaminase levels, and headaches. A smaller proportion of patients may also experience symptoms related to the arthralgia, gastrointestinal tract (such as nausea, vomiting, diarrhea), respiratory system (like cough, pulmonary infiltrates, acute respiratory distress syndrome (ARDS)), liver, or central nervous system [134,135]. The treatment traditionally is based on doxycycline, which is the preferred treatment for HGA in both adults and children over eight years old, demonstrating positive outcomes. Treatment initiation based on clinical suspicion alone is crucial to prevent the risk of serious complications [136]. Other antibiotics, including quinolones, cephalosporins, penicillin, and macrolides, are ineffective against HGA. To prevent infection, it is essential to take precautions to avoid exposure to ticks [137].

Ehrlichiosis in the human is caused by three species: *Ehrlichia chaffeensis*, *E. ewingii*, and a relatively new species identified in 2009 in the upper Midwestern USA, namely, *E. muris* subspecies *eauclairensis* [138]. The main vectors for *E. chaffeensis* and *E. ewingii* are *Amblyomma Americanum* ticks, and the primary natural reservoir host is the white-tailed deer in North America. Other tick vectors, including *H. longicornis* and *R. sanguineus* are involved in transmitting the bacteria to other areas.

Infections caused by *E. chaffeensis*, called human monocytic ehrlichiosis (HME), are more severe than those caused by *E. ewingii* (Human ewingii ehrlichiosis (HEE)). They include typical ehrlichiosis symptoms (fevers, chills, malaise, myalgia, and nausea) but lead to a severe disease, with 60% of cases requiring hospitalization and a fatality rate of 3%. *E. ewingii* mostly affects immunosuppressed individuals [130]. *E. muris* subsp. *eauclairensis* is transmitted by *I. scapularis*, and its suggested host is the white-footed mice. It causes an illness characterized by fever, headache, myalgias, lymphopenia, and thrombocytopenia [138]. Similar to HGA, ehrlichiosis infections are rapidly treated with doxycycline, leading to full recovery.

*Candidatus Neoehrlichia mikurensis*, also known as *Neoehrlichia mikurensis*, is an emerging bacterium identified in the blood of febrile patients in 2010. It can manifest as a recurrent fever accompanied by thromboembolic syndromes, primarily affecting immunosuppressed individuals. This bacterium is commonly found in wild rodents, and two *Ixodes* species (*I. Ricinus* and *I. persulcatis*) are known carriers of the pathogen, detected in ticks across Asia, Russia and Europe [130]. Doxycycline is used to treat this infection [139].

Currently, there are no vaccines available for granulocytic anaplasmosis or ehrlichiosis and related infections (neoerlrichiosis). Since they all are obligate intracellular bacteria, disrupting microbe–host cell interactions that facilitate invasion can potentially disrupt infection.

## 5. Other Tick-Borne Infections

Human babesiosis represents a growing health concern as a tick-borne disease and is caused by a protozoan parasite within the *Babesia* genus, which specifically targets red blood cells. These protozoa are part of the Apicomplexa phylum, which is also home to other well-known pathogens causing malaria (*Plasmodium* spp.) and toxoplasmosis (Toxoplasma gondii) [140]. Globally, over a hundred Babesia species are known to infect a variety of wildlife and domestic animals, but only six have been identified as harmful to humans: these include *Babesia crassa*-like agent, *B. divergens, B. duncani, B. microti, B. motasi,* and *B. venatorum* [141]. Of these, *B. microti and B. duncani* are predominantly found in North America, whereas *B. divergens* is more commonly reported in Europe, marking them as primary agents of human babesiosis. In Asia, *B. venatorum* has recently emerged as a notable pathogen, uniquely impacting those with intact immune systems, unlike European cases where it primarily affects immunocompromised individuals [142]. The primary vector for transmitting *B. microti* is the *I. scapularis* tick, with the white-footed mouse (*Peromyscus leucopus*) serving as the main reservoir in North America [143,144]. *B. duncani*’s primary tick vector is *Dermacentor albipictus* and possibly *I. scapularis*, while suggested reservoirs include birds and the white-footed mouse [141,145]. Although human babesiosis and Lyme disease share common reservoir hosts and tick vectors, the occurrence of babesiosis in humans is notably localized within certain areas where Lyme disease is endemic. Additionally, the incidence of babesiosis in these regions is comparatively lower than that of Lyme disease.

This malaria-like illness often goes unnoticed, likely because most cases are asymptomatic, as suggested by serological surveys. The severity of babesiosis depends mainly on the immune status of the host, the presence of risk factors, and the Babesia species responsible for the infection. When symptoms do manifest, they include fever, chills, sweating, myalgias, fatigue, hepatosplenomegaly, and hemolytic anemia. Typically, these symptoms appear 1 to 4 weeks after the incubation period and can persist for several weeks. The severity of the disease is heightened in individuals who are immunosuppressed, have undergone splenectomy, or are elderly. Infections caused by *B. divergens* and *B. duncani* are typically more severe and can be fatal if not promptly treated, whereas those caused by *B. microti* generally result in clinical recovery [146].

Typically, individuals exhibiting symptoms of babesiosis experience resolution within a week following the commencement of treatment. However, itis noteworthy that persons without symptoms can exhibit low-grade parasitemia that may linger for as long as a year. Presently, there is no available vaccine for human babesiosis. Preventive strategies primarily emphasize avoiding contact with the disease vectors. While not universally successful, the standard treatment approach for babesiosis combines antimicrobial therapy with exchange transfusion. The current treatment options primarily involve four drugs: atovaquone, azithromycin, clindamycin, and quinine. Two widely employed and highly effective treatment combinations include atovaquone with azithromycin and clindamycin with quinine [147]. It is crucial for individuals diagnosed with symptomatic babesiosis to promptly begin antimicrobial therapy.

## 6. Anti-Tick Countermeasures

The battle against tick-borne diseases necessitates an integrated approach in developing and implementing effective anti-tick countermeasures. These strategies can be broadly categorized into chemical, biological, ecological, and immunological methods, each addressing different aspects of tick control and disease prevention.

Chemical Control: Chemical agents, notably acaricides, have been a primary line of defense in controlling tick populations. The development of new chemical compounds, especially those targeting specific tick receptors or biological pathways, offers hope in overcoming resistance issues prevalent in many tick species [148]. Among these, isoxazolines have emerged as a potent class of ectoparasiticides, highly effective in controlling tick infestations in domestic animals and potentially applicable in broader ecological contexts [149]. Despite their efficacy, the environmental impact of these chemicals, including potential effects on non-target species and ecosystem balance, requires careful consideration. Ongoing research into the development of more environmentally friendly and target-specific acaricides is crucial.

Biological Control: The use of natural enemies of ticks, such as entomopathogenic fungi, offers an environmentally sustainable method of control. Species like *Metarhizium anisopliae* and *Beauveria bassiana* have shown effectiveness in reducing tick populations in controlled studies [150]. However, the application of these fungi in natural settings presents challenges, including ensuring their survival and spread in diverse ecological niches. Furthermore, research into genetically modifying these organisms to enhance their effectiveness against ticks is an exciting frontier, though it comes with ethical and environmental considerations [151].

Ecological Management: Altering the environment to make it less conducive for ticks involves strategies like vegetation management, controlled burns, and wildlife population control. Habitat modification, such as reducing leaf litter and brush where ticks thrive, can significantly reduce tick encounters [152]. Additionally, the use of acaricide-treated materials in wildlife habitats, like the aforementioned “tick tubes”, targets the immature stages of ticks, breaking the life cycle and reducing overall populations. Public education emphasizing personal protection measures, such as the use of repellents and appropriate clothing—including the utilization of permethrin-treated clothes—constitutes a vital component of ecological management [153]. 

Integrated Tick Management (ITM): The most effective strategy against tick-borne diseases is likely to be a comprehensive approach that combines chemical, biological, ecological, and immunological methods. ITM involves coordinated efforts between public health officials, researchers, and communities to implement strategies tailored to specific ecological and sociocultural contexts [154]. Monitoring and surveillance play a critical role in ITM, enabling the timely adjustment of strategies in response to changing tick populations and disease patterns. Community engagement and education are also critical, ensuring public awareness and participation in tick control efforts [155].

Vaccination: The development of vaccines against ticks represents a significant advancement in tick control. Current research focuses on identifying and targeting key proteins involved in tick feeding and pathogen transmission [156]. These vaccines aim not only to protect the host from tick bites but also to interrupt the transmission of tick-borne pathogens, a concept known as “transmission-blocking” [6]. The development of such vaccines requires an in-depth understanding of tick biology, host immune responses, and the Intricate interactions between ticks, pathogens, and hosts.

In summary, the control of tick populations and the prevention of tick-borne diseases requires a multifaceted and integrated approach. Continuous research, innovation, and collaboration across various disciplines are crucial in developing effective and sustainable anti-tick strategies. The advancement of technologies, particularly in the field of vaccine development and biological control, holds great promise for future endeavors in this domain.

In the next section, we provide an overview of the latest advancements in antigen-based vaccine development, including a brief summary of emerging antigens proposed as anti-tick vaccine candidates. This encompasses information about their sources and the methodologies utilized to assess their efficacy.

## 7. Advances in Anti-Tick Vaccines

The global economic burden in agriculture and healthcare, attributed to ticks and the diseases they transmit, results in estimated annual losses of up to USD 30 billion [157]. As previously mentioned, the preferred strategy for tick control involves integrating multiple approaches. These encompass a variety of methods, including the development of host resistance, management of tick abundance, biological control, acaricide resistance management, and developing vaccines against ticks. While tick-borne diseases undoubtedly impact both farm and domestic animals, vaccination holds the potential not only to safeguard these animals but also to have a significant impact on humans. By targeting reservoir hosts, including wild and farm animals, with veterinary vaccines, we can not only benefit animal health and agriculture but also significantly decrease the risk of transmission of various tick-borne diseases to humans.

The progress in genomics and other “omics” fields over the past two decades has given rise to a “third generation” of vaccines. These vaccines are developed using cutting-edge technologies like functional omics, reverse vaccinology, and the systems biology approach. This advancement aims to address the limitations of traditional vaccine development methods. In this evolving approach, vaccine development has become more customized, focusing on the specific antigen components targeted by protective immune responses. Researchers adopt a comprehensive viewpoint, considering both the pathogen and its interaction with the host immune system. This tailored strategy enhances the effectiveness and precision of vaccine development efforts. Following the pivotal research by Allen and Humphreys in 1979 [158], a plethora of studies have emerged [159,160]. These investigations have employed a diverse spectrum of antigens, ranging from complete tick homogenates to specific internal organs, aiming to develop various degrees of immunity against tick infestations [161]. The approach of vaccination or immunological control stands out as the most promising, eco-friendly, and sustainable method in combating ixodid tick infestations. To date, a significant number of antigens have been identified that offer protection against these ticks. The range of effective immune responses against argasid ticks is notably narrower compared to those for ixodid ticks [162,163,164,165]. Trager’s early work highlighted that guinea pigs and rabbits could develop immune resistance to ixodid tick larvae after repeated exposure [166]. This was further evidenced when guinea pigs demonstrated a similar immune response upon being inoculated with native protein tissue extracts from *Dermacentor variabilis* ticks. Trager’s 1940 study extended these findings to argasid ticks, observing that chickens developed a partial immunity after exposure. This immunity was characterized by reduced tick feeding, lower egg weight, and decreased egg viability in the ticks [167,168,169]. Additionally, several research groups have reported that host antibodies, particularly immunoglobulins (IgGs), are capable of penetrating the gut epithelium of ticks and reaching their hemolymph and other internal tissues [170,171,172].

Moreover, antibodies developed against specific tick vaccine antigens have exhibited reactivity against corresponding tick tissue proteins [173]. This suggests that sera containing anti-tick antigens, when ingested during blood feeding, might disrupt the normal functioning of internal tick proteins. Building upon these findings, various recombinant tick antigens have been identified against ixodid ticks [165,174]. Among these, Bm86 stands out as the most successful in field conditions. This glycoprotein derived from the digestive tract of *Rhipicephalus microplus* is the basis for the only commercially available anti-tick vaccine (Gavac^®^), widely used to protect cattle against multiple tick infestations in numerous countries [175]. Nonetheless, researchers in tick biology are persistently striving to develop effective vaccines capable of eliciting robust humoral or cell-mediated protective immune responses in both argasid and ixodid ticks. However, accomplishing this objective involves more than just creating an effective anti-tick vaccine.

At present, the development of new anti-tick vaccines involves the crucial task of identifying and characterizing novel tick antigen candidates, as well as evaluating different combinations of antigens as a cocktail approach [176]. Ideally, these candidates should be highly conserved and capable of inducing cross-reactive immunity against multiple tick species throughout all stages of the tick life cycle in the targeted hosts.

### 7.1. Mechanisms and Targets of Anti-Tick Vaccines

Developing effective anti-tick vaccines hinges on the successful identification of specific antigens. A thorough comprehension of the interactions between host and pathogen at the molecular level is critical for this process. This knowledge is key to pinpointing potential antigen targets. Recent studies have shown that synthetic and recombinant proteins can induce some level of protective immunity [177]. This section highlights the most promising antigen candidates derived from various tissues, which have been rigorously tested and are considered viable for anti-tick vaccine development, according to current research. A list of the antigens is presented in Table 2, including antigens known to have been tested for immunogenicity in animals, resulting in a certain level of protection efficacy against ticks or a decrease in the transmission of tick-borne pathogens.

#### 7.1.1. Tick Salivary Gland Antigen Candidates

Targeting tick antigens has emerged as a promising strategy by which to control tick infestations and prevent the transmission of tick-borne diseases. Several tick salivary and other associated proteins have been identified as novel potential vaccine candidates, each playing a unique role in the tick–host–pathogen interactions.

A recent study has identified salivary protease inhibitors as potential antigens for anti-tick vaccines. Three protein candidates, namely, AsKunitz, As8.9kDa, and AsBasicTail, derived from *Amblyomma sculptum*, the primary vector of *Rickettsia rickettsi* in Brazil, were used for immunization of mice. This resulted in a high percentage of efficacy against female ticks (up to 85%), and the mortality rate of nymphs feeding on immunized mice reached 70–100% [178].

Cystatins are found in both soft and hard ticks, exhibiting limited amino acid sequence similarity with other organisms, typically less than 40%. They likely play a role in blood digestion with *R. microplus* cystatin (Bmcystatin), known for its involvement in embryogenesis. Various cystatin orthologs were evaluated for their immunogenicity and protective effects. This evaluation included *I. scapularis* Sialostatin2, *O. moubata* Omc2, and *R. microplus* Bmcys2c, where these immunizations primarily impacted tick engorgement and feeding ability [179,180,181,182].

The salivary glands of ixodid ticks contain metalloproteases (MPs) and are pivotal in tick physiology, serving as multifunctional proteins engaged in a wide array of complex physiological and pathological processes in living organisms [183]. Their presence across different tick species underscores their crucial role in maintaining essential functions associated with blood meals in ticks [184,185]. In other studies, the recombinant *R. microplus* MP proteins have demonstrated substantial efficacy in reducing tick feeding and reproduction, egg production, and hatching rates, positioning them as an interesting candidate for the development of an anti-tick vaccine [186].

**Table 2 vaccines-12-00141-t002:** Antigen candidates and their efficacy in protecting against tick infestation and/or pathogen transmission.

Antigen Candidates	Formats	Species	Efficacy on Ticks	Efficacy on Pathogen Transmission	Ref.
**Tick Salivary Gland and Saliva associated antigens**					
AsKunitz, As8.9kDa and AsBasicTail	rProtein	*A. sculptum*	Up to 85% overall efficacy against female ticks; mortality of nymphs of 70–100%	N/D	[178]
Cystatins	Sialostatin 2 (rProtein)	*I. scapularis*	Reduction of feeding ability	N/D	[179]
	Omc2 (rProtein)	*O. moubata*	Reduction of tick survival and engorgement		[182]
	rBmcys2c (rProtein)	*R. microplus*	Reduction of engorgement for *R. appendiculatus*; no effect against *I. persulcatus*		[180,181]
Isac	Adenoviral vector	*I. scapularis*	N/D	*Borrelia burgdorferi* (40% protection) when combined with Salp15	[187]
Metalloproteases	BrRm-MP4 (rProtein)	*R. microplus*	Overall efficacy of 60%	N/D	[186]
	rHLMP1 (rProtein)	*H. longicornis*	Mortality of nymphs and adult ticks (15.6% and 14.6%, respectively)		[188]
Ribosomal p0	rP0 (rProtein), synthetic peptide	*Rhipicephalus* spp.	Overall efficacy of 90% against *R. sanguineus*	N/D	[189,190]
	Synthetic peptide		Overall efficacy of 96% and 54% against *R. microplus* and *A. mixtum*, respectively		[191,192]
	Synthetic peptide conjugated to Bm86 rProtein		Overall efficacy of 63% and 55% against *I. ricinus* and *D. nitens*, respectively		[193]
Salp15	rProtein, Adenoviral vector, DNA	*I. scapularis*	Reduction in tick weight (rProtein)	*Borrelia burgdorferi* (40–60% protection) when combined with Isac (Adeno) or with rProtein	[187,194,195]
Serpins	BmTI (rProtein)	*R. microplus*	Overall efficacy of 72.8%	N/D	[196]
	RmLTI (rProtein)		Overall efficacy of 32%		[197]
	RmS-17 (peptide)		Overall efficacy of 79%		[198]
	rHLS2 (rProtein)	*H. longicornis*	Mortality of nymphs and adult ticks (44.6% and 43%, respectively)		[199]
	Iris (rProtein)	*I. ricinus*	Reduction in tick weight and mortality of 30%, only in rabbit and not mice		[200]
	RAS-3, RAS-4, RIM36 (rProtein)	*R. appendiculatus*	39.5 mortality of female ticks on vaccinated animals	Reduction of *Theileria parva* infection (38%)	[201]
SILK	rProtein	*R. microplus*	Overall efficacy of 62%	*Anaplasma marginale* (less DNA in vaccinated cattle)	[202]
SUB	rProtein	*R. microplus*	Overall efficacy of 86.3% against *Rhipicephalus sanguineus*	N/D	[190]
			N/D	Reduction of *A. marginale* (98%) and *Babesia begemina* infections (99%)	[203]
			Overall efficacy of 60%	*A. marginale* and *B. begemina* (less DNA in vaccinated cattle)	[202]
			Overall efficacy of 51% and 60% against *R. microplus* and *R. annulatus*, respectively	N/D	[204]
		*I. scapularis*	N/D	Reduction of *A. phacocytophilum* infection (33%)	[205]
	Vaccinia virus vector		Reduction of tick infestation (52%)	Reduction of *B. burgdorferi* infection (40%)	[206]
	rProtein	*H. anatolicum*	Overall efficacy of 65.4% and 54% against *H. anatolicum* and *R. microplus*, respectively	N/D	[207]
tHRF	rProtein, DNA	*I. scapularis*	Reduction of engorgement with rProtein t	Reduction of *B. burgdorferi* transmission with rProtein	[194,208]
TSLPI					[194,209]
TIX-5					
UBQ	Synthetic peptide	*R. microplus*	Overall efficacy of 55% and 15% against *R. microplus* and *R. annulatus*, respectively	N/D	[204]
**Gut-associated antigens**					
AQPs	RmAQP1 (rProtein)	*R. microplus*	Overall efficacy of 68–75%	N/D	[210]
	RmAQP2 (synthetic peptides)		Overall efficacy of 25%		[211]
	OeAQP/OeAQP1	*O. erraticus*	Overall efficacy of 15.5% and 4.6% against *O. erraticus* and *O. moubata*, respectively	N/D	[212]
ATAQ	Synthetic peptide	*R. microplus*	Overall efficacy of 35% and 47% against *R. microplus* and *R. sanguineus*, respectively	N/D	[213]
Ba86	rProtein	*R. annulatus*	Overall efficacy of 71.5% and 83% against *R. microplus* and *R. annulatus*, respectively	N/D	[214]
Bm86 and derived antigens	rProtein	*R. microplus*	Reduction of tick weight and egg-laying capacity	Overall reduction of animal deaths caused by *Babesia* spp. (76%)	[215]
			Overall efficacy of 51%	N/D	[216]
	Bm4912 (Synthetic peptide)		Overall efficacy of 72.4%		
	Bm19733 (Synthetic peptide)		Overall efficacy of 35.87%		
	Bm7462 (Synthetic peptide)		Overall efficacy of 72.4% to 81.05%		[216,217]
Bm95	rProtein	*R. microplus*	Overall efficacy of 58% and 89% against *R. microplus* strains A and Camcord, respectively	N/D	[218]
			Overall efficacy of 81.27%		[219]
FER1/FER2	HlFER1 (rProtein)	*H. longicornis*	Overall efficacy of 34%	N/D	[220]
	HlFER2 (rProtein)		Overall efficacy of 49%		
	Ip-FER2	*I. persulcatus*	Reduction of tick engorgement in *I. persulcatus* and *I. ovatus*		[221]
	FER2 (rProtein)	*H. anatolicum*	Overall efficacy of 95.9% against *Rhipicephalus sanguineus*		[190]
	HaFER2 (rProtein)		Protection against larvae and adult ticks (51.7% and 51.2%, respectively)		[222]
	RmFER2 (rProtein)	*R. microplus*	Overall efficacy of 64% and 72% against *R. microplus* and *R. annulatus*, respectively		[223]
	IrFER2 (rProtein)	*I. ricinus*	Overall efficacy of 98%		
GST	GST-Hl (rProtein)	*H. longicornis*	Overall efficacy of 57% against *R. microplus*	N/D	[224]
Haa86	rProtein	*H. a. anatolicum*	Reduction of tick feeding, weight and overall efficacy of 82%	60% survival to *Theileria annulata* infection lethal challenge	[225]
			Protection against larvae and adult ticks (68.7% and 45.8%, respectively); overall efficacy of 36.5% against *R. microplus*	N/D	[226]
Is86	Is86 EGF-like domains (rProteins)	*I. scapularis*	Reduction of tick engorgement	Reduction of B. borrelia infection	[227]
5′nucleotidase	rProtein	*R. microplus*	Reduction in tick engorgement	N/D	[228]
TROSPA	rProtein	*R. microplus*	No effect	*B. bigemina* (less DNA in vaccinated cattle)	[202]
**Cement-associated antigens**					
64TRP	rProtein	*I. ricinus*	N/D	Up to 71% survival to TBEV infection lethal challenge	[229]
	Truncated 64TRP (rProteins)	*R. appendiculatus*	Protection against *R. sanguineus* and *I. scapularis* ticks	N/D	[230]
**Egg-associated antigens**					
BYC	rProtein	*R. microplus*	Overall efficacy of 25.24%	N/D	[231]
	Native protein		Overall efficacy of 14–36%		[232]
GP80			Overall efficacy of 68%		[233]
Vitellin					
VTDCE			Overall efficacy of 21%		[234]
**Other antigens**					
ABC transporter	OeABC (rProtein)	*O. erraticus*	Overall efficacy of 3.1% and 22.5% against *O. erraticus* and *O. moubata*, respectively	N/D	[212]
Selenoproteins	OeSEL (rProtein)		Overall efficacy of 47.5% and 9.6% against *O. erraticus* and *O. moubata*, respectively		
Calreticulin	rProtein	*H. anatolicum*	Overall efficacy of 41.3% and 37.6% against *H. anatolicum* and *R. microplus*, respectively	N/D	[207]
Cathepsin L-like cysteine proteinase			Overall efficacy of 30.2% and 22.2% against *H. anatolicum* and *R. microplus*, respectively		

rProtein: recombinant protein.

Another vaccine candidate, the ribosomal protein P0, plays a significant role in regulating ribosomal translational activity, enabling ticks to adapt to various environmental conditions. Tick saliva was shown to contain ribosomal proteins that aid in evading the host innate immunity. It is integral to the formation of a crucial stalk-like structure in the largest ribosomal subunit. Its phosphorylation capacity is essential for modulating the ribosome’s translational activity. P0’s interaction with P1, P2, 28S rRNA, and the eEF2 factor is critical; its absence can lead to defective 60S ribosomal subunits, halted protein synthesis, and consequent cell death [189]. In vaccine research, P0 has emerged as a promising candidate. In a study involving rabbits, a vaccine developed from a synthetic peptide containing 20 amino acids of this sequence demonstrated efficacy in protecting against *Rhipicephalus sanguineus* infestations during an immunization and challenge experiment. Furthermore, a study that created a 20-amino-acid peptide derived from the P0 protein of Rhipicephalus ticks, linked with keyhole limpet hemocyanin (KLH), showed impressive results. This P0-KLH combination was 96% effective against *R. microplus* in cattle [191,192]. While the promise of P0-KLH as a vaccine component is clear, its high production costs could limit its practicality for large-scale use in livestock. Further research on developing an antigenic vaccine using recombinant methods is essential to assess efficacy and improve economic viability. Researchers are exploring the combination of these antigens into multi-antigen constructs, leading to the development of chimeric proteins with enhanced efficacy.

Salp15, an immune-suppressive salivary protein found in *Ixodes scapularis*, actively inhibits CD4+ T cell activation, suppresses complement activity, reduces cytokine production, and hampers dendritic cell function in the host [235,236]. It is primarily involved in *Borrelia burgdorferi* pathogenicity and transmission by interacting with the outer surface protein C (OspC) of the bacterium. Blocking this interaction could potentially reduce the transmission of the pathogen to vaccinated hosts. Salp15 can be produced in significant quantities as recombinant proteins using the *Escherichia coli* expression system, an important feature for cost-effective vaccine production [236,237]. Furthermore, an immunization combining Salp15 and Isac, a protein from *I. scapularis* known for its ability to inhibit complement activity in the host, has demonstrated capacity to reduce *B. burgdorferi* transmission and provide 40% protection in immunized mice [187].

Serine protease inhibitors (serpins) have been isolated from various tick species and exhibit antigenic properties. These serpins play significant roles in animal physiology, notably in cattle, where they influence blood clotting by modifying prothrombin time and partially initiating thromboplastin time [238,239]. Additionally, serpins assist ticks in their initial feeding process by subduing the immune system responses of the tick [240]. A study using *R. microplus* trypsin inhibitors (BmTIs) demonstrated promising results with 72.8% efficacy against *R. micropulus*; however, further research revealed that truncated or whole recombinant protein were less effective [197], underscoring the importance of a correctly folded and post-translational states of the recombinant protein in successful immunization. Other serpins have recently been suggested as potential antigen candidates, such as the *I. scapularis* nymph tick saliva serpin IxsS41, which is implicated in tick evasion of host immunity and *Borrelia burgdorferi* pathogenesis. Further studies are required to confirm its ability to protect against ticks or pathogen transmission [241].

Following transcriptomic analyses of salivary glands, further investigations into the flagelliform silk proteinSILK, sourced from *R. microplus* and *Dermacentor andersoni*, have revealed notable protective effects against both tick infestations and tick-borne infections in cattle [202].

Furthermore, proteins like subolesin (SUB), the tick ortholog of vertebrate Akirin, have been extensively explored as potential vaccine candidates. The primary regulatory role of Subolesin/Akirin involves mediating protein–protein interactions with chromatin remodelers, transcription factors, histone acetyltransferases, RNA-associated proteins, and importins, possibly implicating direct interactions with chromatin [242,243]. SUB, found across various tick species, has shown promise in vaccination efforts, such as in cattle, conferring protective effects against multiple ticks and pathogen transmission [243,244,245]. Vaccines utilizing SUB, either alone or combined with other antigens, have demonstrated efficacy ranging from 80% to 97%, comparable to or surpassing that of well-known tick antigens like Bm86, metalloprotease, ribosomal protein P0, ferritin 2, and aquaporin [245].

*Ixodes scapularis* tick histamine release factor (tHRH), tick salivary lectin pathway inhibitor (TSLPI) and tick inhibitor of factor Xa toward factor V (TIX-5) have demonstrated the ability to reduce engorgement in vaccinated cattle and *B. borrelia* transmission in rabbit immunized with the recombinant proteins. However, a similar experiment using DNA did not offer protection or prevent pathogen transmission, emphasizing the crucial role of the antigen format used [194,197,209].

Ubiquitin (UBQ) is a ubiquitous protein involved in several cellular processes and post-translational modifications. A study involving the vaccination of cattle with a recombinant peptide of *R. microplus* conferred protection against *R. microplus* (55%), with a lesser cross-effect on *R. annulatus* (15%), suggesting that this antigen is more likely to be species-specific [204].

These findings underscore the importance of understanding tick salivary proteins and their interactions with hosts and pathogens. Targeting salivary gland-associated antigens presents a viable approach for developing effective anti-tick vaccines, mitigating the impact of tick infestations on livestock and public health.

#### 7.1.2. Tick Midgut Antigen Candidates

Ticks also require specialized proteins in their midgut to facilitate vital physiological processes. Among these, aquaporins (AQPs), vital in water homeostasis and cryoprotection, are highly conserved transmembrane proteins that form channels facilitating water and solute transport across cell membranes [246]. AQPs in ticks have been found in organs like the digestive tract, Malpighian tubules, and salivary glands, controlling blood volume intake during feeding [247]. They significantly reduce blood volume in tick guts, which is crucial as ticks consume substantial blood relative to their size. A specific aquaporin fragment from engorged female *R. microplus* ticks has been isolated, recombinantly synthesized, and identified as the RmAQP1 vaccine [210]. Trials demonstrated up to 75% efficacy in vaccinated cattle. RmAQP2 has shown potential, reducing tick feeding by 25% [211]. These findings suggest that targeting AQPs could disrupt the ability of the ticks to maintain water balance, impacting their feeding capabilities and overall survival.

Reverse vaccinology approaches allowed for the evaluation of a new antigen candidate, namely, a synthetic peptide from the ATAQ protein of *R. microplus*. This protein is present in the gut and Malpighian tubes of different tick species and was tested for its efficacy in mice, rabbits, and cattle, displaying efficiency against *R. microplus* and *R. sanguineus* infestation. [213].

Building on the success of the Bm86-based anti-tick vaccine, research involving different tick Bm86 orthologs or derived antigens has been conducted. Examples include immunogenicity experiments that have demonstrated success with recombinant proteins like Ba86 against *R. annulatus* and *R. microplus*, Haa86 against *H. a. anatolicum* and *R. microplus* infestation, as well as some protection against *Theileria annulata* infection [214,225,226]. Additionally, a recently explored ortholog from Ixodes scapularis, known as Is86, has shown efficacy in reducing *Borrelia burgdorferi* spirochete load in the skin. It has shown influence on tick engorgement and molting in vaccinated mice. Although immunization with the current recombinant protein formats—specifically, three distinct regions carrying epidermal growth factors (EGF)-like domains of Is86 (Is861, -2 and -3)—exhibited a low but significant impact on tick engorgement and pathogen survival, further studies are required to confirm the potential of Is86 as an anti-tick vaccine antigen [227]. Other synthetic peptides derived from Bm86 have also shown some efficacy against *R. microplus* tick infestation [216,217].

Another midgut protein from *R. microplus*, Bm95, was extensively studied initially in response to the lack of efficiency of the commercially available Bm86-based vaccine against specific strains. Bm95 demonstrated interesting protective effects against multiple strains, suggesting a more universal antigen [218,219].

Ferritin proteins, namely, FER1 and FER2, are essential for iron metabolism, allowing ticks to process ingested blood efficiently [248]. FER2, primarily expressed in the gut, has emerged as a promising vaccine candidate due to its role in iron transport and its impact on tick feeding, oviposition, and larval hatching. In addition to FER2, FER1 has also been identified as a viable antigen candidate for controlling various tick species. Studies were conducted on the recombinant FER2 protein of *R. microplus* (RmFER2) in cattle immunization, revealing that the FER2-based vaccine exhibited an overall efficacy of 64% [249]. Most recently, another study highlighted the strong protection observed in a calf vaccinated with *Hyalomma anatolicum* FER2, a vector for CCHFV. The vaccination exhibited robust defense against both larval (51.7%) and adult (51.2%) tick infestations [222]. Several recent studies have consistently affirmed the substantial protective effect of FER2 against tick infestation, utilizing recombinant FER2 protein [250,251]. In a recent investigation, researchers identified FER2 orthologues in soft ticks *Ornithodoros moubata* (OMFER2) and *Ornithodoros erraticus* (OEFer2), revealing an 85.3% sequence similarity. While the recombinant form (tOMFER2) induced robust immune responses in rabbits, it failed to exhibit protective effects against *O. erraticus*, despite its close sequence match. This disparity suggests that subtle sequence differences might dictate their effectiveness. Nevertheless, the study highlights the potential of OMFER2 as a promising antigen candidate for vaccine development [252].

Glutathione S-transferases (GSTs), previously identified as ligandins, represent a multifunctional protein family prevalent across various animal species. These enzymes are involved in diverse biological functions such as intracellular transport, digestion, prostaglandin production, detoxification of internal and external substances, and counteracting oxidative stress. Increased levels of GST expression occur in organisms exposed to insecticides and acaricides [253]. In other studies, GSTs have shown partial cross-protective immunity in hosts when used in vaccines against *Rhipicephalus* and *Haemaphysalis* ticks. This suggests that the protective ability of GST alone might not be adequate and may exhibit limited efficacy as a single-antigen vaccine against multiple tick infestations [224,254].

An ectoenzyme primarily located in the Malpighian tubules of *R. microplus*, known as 5′-nucleotidase, was evaluated for its potential as a vaccine in sheep and cattle. Vaccinated sheep exhibited higher titers of anti-nucleotidase antibodies and significant protection against adult ticks, leading to a notable reduction in tick engorgement. However, no protection was observed in cattle. This underscores the intricacy of the interaction between the antigen and the host, emphasizing the specific requirements for different host species [228].

Another significant target is TROSPA, a tick receptor crucial for *Borrelia burgdorferi* spirochete colonization in *I. scapularis* ticks. Inhibition of the interaction between TROSPA and spirochetes could hinder pathogen transmission [255,256]. While research into TROSPA-based vaccines has shown promise, further studies are essential to optimize its effectiveness and explore its application across different tick species.

#### 7.1.3. Tick Cement Antigen Candidates

Tick cement, a blend of glyco- and lipoproteins secreted into the host during tick attachment, serves as a valuable reservoir of tick-derived antigens for vaccine development [257]. Apart from securing tick mouthparts to the host skin, tick cement also acts as a storage site for pathogens like *B. burgdorferi* sensu lato and TBEV [258,259]. Interesting antigens have been identified from tick cement, demonstrating effectiveness in controlling tick infestation and tick-borne diseases. One notable antigen, 64P (64TRPs), a 15 kDa cement protein secreted by *Rhipicephalus appendiculatus* salivary glands, has shown cross-protection against different tick species, including *Rhipicephalus sanguineus* and *Ixodes ricinus*. Furthermore, truncated constructs of 64P significantly reduced nymphal and adult tick infestations, causing up to 70% adult tick mortality, also impairing attachment and feeding. This broad-spectrum vaccine antigen has proven effective against various tick stages and species. It operates by inhibiting tick attachment and feeding while also interacting with concealed midgut antigens, leading to engorged tick mortality [230]. The 64P antigen not only enhances antibody responses but also exhibits cross-reactivity with different tick tissues, offering advantages from both “concealed” and “exposed” antigens [260]. These findings highlight the potential of tick-cement-associated antigens in developing vaccines for effective tick control.

#### 7.1.4. Egg-Associated Antigen Candidates

Proteins found in tick eggs were also investigated and tested for immunogenicity and protection. Immunization with *Boophilus* (*Rhipicephalus*) yolk pro-cathepsin (BYC) in both its native and recombinant forms resulted in protection efficacy of 14.36 and 25.24% against *R. microplus*, respectively [231,232]. The most abundant protein present in *R. microplus* eggs is vitellin, derived from a large precursor called vitellogenin. A processed product from this precursor is a glycoprotein named GP80. Both isolated native proteins were evaluated in sheep vaccination and resulted in fewer engorged female ticks and an overall protection rate of 68% against *R. microplus* [233]. Vitellin-degrading cysteine endopeptidase (VTDCE) is an enzyme involved in *R. microplus* embryogenesis and was tested for its vaccine potential, demonstrating an overall efficacy of 21% against *R. microplus* infestation [234]. While egg-associated antigen candidates showed some level of protection, combining them with other types of antigens may be necessary to enhance overall protection.

#### 7.1.5. Other Antigen Candidates

A study based on the midgut transcriptomic and proteomic data of the soft tick *O. erraticus* allowed for the identification of novel antigen candidates belonging to functional protein families that have never been explored so far, namely, an ABC transporter and a Selenoprotein T. Efficacy was evaluated against two tick species, *O. erraticus* and *O. moubata*, resulting in some cross-reactivity and protection rates ranging from 3.1% to 47.5% [212].

Two proteins from *H. anatolicum*, calreticulin and cathepsin L-like cysteine protease, were evaluated for cross-reactivity against *R. microplus*. Both demonstrated protective effects against both ticks, with calreticulin exhibiting a higher degree of protection [207].

### 7.2. Types of Anti-Tick Vaccines

Ticks transmit diseases to both humans and livestock, contributing to over 60% of zoonotic infections in humans [261]. Livestock farmers face substantial economic losses due to tick-borne diseases, compelling the need for effective tick control methods [262]. Vaccines offer a safer and cost-effective approach, although they often target specific tick species. Anti-tick vaccines aim to diminish tick populations and the prevalence of tick-borne diseases by targeting tick feeding, reproduction, and development. This is achieved through antigen-specific antibodies that interfere with tick protein functionality and other immune mechanisms. Research programs worldwide are focused on developing vaccines that are effective against multiple tick species. Strategies involving the combination of unrelated antigens within a single vaccine hold the potential to enhance cross-protection against multiple tick species. However, many types of vaccines are now available, and depending on the strategy, some could be favored in terms of cost, route of administration, immunogenicity, long-term stability, safety, and antigen capacity.

#### 7.2.1. Protein-Based Vaccines

Generating immunity via vaccination through protein antigens has been a focal point in immunology and drug delivery. Protein vaccines leverage peptides or proteins as antigens to trigger an immune response. These antigens can originate from the pathogen causing the disease or from other sources such as the ticks, sourced naturally in their complete form or as fragments.

Numerous studies have focused on protein-based vaccines to combat ticks and tick-borne diseases. These endeavors began in the 1970s with experimental formulations targeting tick species like *D. andersoni* [158]. The initial studies assessing vaccine efficacy encouraged global research, particularly in the 1980s, investigating immune responses in bovines against *R. microplus* [263]. The Bm86 recombinant protein, produced via a yeast expression system, contributed to commercial vaccines such as TickGARD^®^ (now unavailable) and Gavac^®^. These vaccines notably decreased tick populations by up to 74%, demonstrating diverse efficacy levels ranging from 51% to 91% [264,265,266,267]. However, variations in tick populations, especially in regions like Colombia, Mexico, Brazil, and Argentina, displayed differing responses to the Bm86 vaccine due to polymorphisms in the Bm86 antigen genes [218,268]. To enhance Bm86 vaccine effectiveness, peptides derived from Bm86, such as SBm4912, SBm7462^®^, and SBm19733, were developed, demonstrating efficacy ranging from 35.87% to 81.05% [216,217]. Recent efforts, like the Bovimune Ixovac^®^ vaccine, aim to boost Bm86 vaccine effectiveness, but their efficacy remains unverified against ticks [269]. Despite the success of the Bm86 vaccine against *R. microplus* ticks, its applicability to other tick species like *Ixodes* remains uncertain due to differing physiological and genetic factors [164,177,270,271]. While the Bm86 vaccine has proven effective, its inability to eliminate ticks entirely limits its standalone use, yet it significantly reduces the need for acaricides, lessening the economic burden of tick-borne diseases in livestock [272,273].

These findings prove cost-effective in mitigating tick infestations and associated diseases, exhibiting promising potential for livestock management. However, despite the identification of various protein antigens capable of inducing immunity, the advancement of protein-based vaccines faces challenges, primarily related to issues of efficient production. Most importantly, there is the potential issue of the antigen lacking correct folding or having an improper glycosylation state—crucial features for optimal immunogenicity.

#### 7.2.2. DNA-Based Vaccines

Vaccination remains one of the most effective applications of immunology in safeguarding human health. Regular reviews of vaccine efficacy and stringent safety evaluations are paramount. DNA vaccines, employing a straightforward yet potent method, stimulate comprehensive immune responses. They achieve this by enabling the expression of microbial antigens within host cells that acquire the plasmid. This process generates the target antigen internally, potentially aiding presentation through the major histocompatibility complex [274].

Many research groups have extensively explored DNA vaccines to control tick infestation. These vaccines, distinct from traditional protein-based ones, encode antigenic proteins within bacterial plasmids controlled by eukaryotic promoters [275]. DNA vaccines offer a simple design, high stability, and safe administration. They introduce plasmid DNA into cells, residing as episomal DNA within the nucleus to continuously generate protective antigens throughout the cell’s lifespan [276]. This approach mitigates the need for repeated boosting, as antigens are continuously produced in vivo, stimulating both cellular and humoral immune responses. The expressed antigen can be presented by MHC class I and II complexes, thereby stimulating CD4+ and CD8+ T cells [277]. Moreover, as DNA vaccines lack contaminating proteins, multiple administrations are unlikely to trigger immune reactions to the vector DNA [278]. Recent advancements in DNA vaccine technology have shown promise in preventing tick infestations [279,280,281]. These vaccines, which are different from traditional protein-based ones, use bacterial plasmids encoding antigenic proteins, offering advantages like simplicity, stability, and safety [282]. Key research includes Sayed et al.’s study [282], where DNA from *Argas persicus* eggs used to immunize chickens reduced tick feeding by up to 89.39%. Another significant finding was observed in BALB/c mice, where the pBMC2 plasmid encoding the Bm86 antigen led to a stronger immune response against *R. microplus*. This was evident through increased anti-Bm86 antibodies, higher interleukin and IFN-γ levels, and stimulated splenocytes. 

Similarly, rabbits vaccinated with a paramyosin (Pmy) DNA vaccine showed a high IgG response, partially protecting against *Hyalomma longicornis* infection. A multi-epitope DNA vaccine provided 100% protection to sheep against *Ehrlichia ruminantium* under lab conditions but not in the field. Other antigens like Salp14 and lipocalins were also evaluated as DNA vaccines. Salp14 DNA vaccination caused erythema at the tick bite site, while a lipocalins (LIP) vaccine from *H. longicornis* influenced tick engorgement weight, oviposition, and hatchability but provided partial protection to the host, indicating limited efficacy [280,283]. Overall, the primary goal of DNA vaccine development should be to drive the host immune response toward Th2, crucial in tick immunity. Cattle vaccinated with *R. microplus* midgut antigens showed elevated specific IgG1 levels, correlating with protection. Optimizing the vaccine design to encourage a Th2 response and the secretion of target antigens into the extracellular space could enhance humoral immunity. Coinfecting Th2 cells with immunomodulatory genes like IL4 and IL10 might favor Th2 cell selection.

#### 7.2.3. mRNA-Based Vaccines

mRNA vaccines signify a shift from traditional vaccination methods by harnessing nucleic acids to prompt the production of microbial antigens within tissues. Unlike classical vaccines that introduce pre-existing antigens, mRNA vaccines operate via a fundamentally distinct approach. These vaccines stem from the advancements in molecular biology that followed the revelations about the inheritance role of nucleic acids, coupled with the subsequent elucidation of DNA structure a decade later [284,285]. The search for effective antigens for anti-tick vaccines has seen significant progress in recent times, using recombinant proteins as a favored platform for testing their efficacy in model organisms. An mRNA vaccine encoding a combination of tick salivary proteins showed promising results in guinea pigs, significantly reducing the transmission of *Borrelia burgdorferi*, the causative agent of Lyme disease. However, despite eliciting immune responses, robust tick immunity through vaccination remains elusive, possibly due to the dynamic alteration of tick salivary proteins during different feeding stages, as suggested by recent research on *Ixodes scapularis* salivary proteins [286]. In a study focused on *I. scapularis*, researchers identified and selected 19 salivary proteins, engineering nucleoside-modified mRNAs encoding these proteins encapsulated in lipid nanoparticles (LNPs). Immunizing guinea pigs with this mRNA-LNP vaccine resulted in robust antibody responses against ten antigens and altered feeding behavior in *I. scapularis* nymphs. Vaccinated animals developed significant erythema and poor tick feeding, leading to detachment and averted *B. burgdorferi* infection. Gene expression analysis indicated the activation of multiple immune pathways, suggesting T cell responses stimulated by the vaccine [286]. In a parallel study, the same researchers investigated Salp14, using mRNA-LNP, plasmid DNA, or recombinant protein platforms for vaccination [283]. Salp14 mRNA immunization induced the most robust humoral response compared to DNA and protein vaccination. Guinea pigs immunized with Salp14 mRNA displayed intense erythema at the bite site; however, this did not influence tick detachment or engorgement weights. Though erythema is considered crucial for tick vaccines, altering later feeding stages might necessitate the inclusion of multiple salivary tick antigens. These findings suggest that Salp14, especially when used as an mRNA-LNP vaccine, holds promise as a candidate for tick vaccines, potentially in combination with other antigens [286].

Overall, the multivalent mRNA vaccine demonstrated promising potential in inducing tick resistance and preventing tick-borne infections in laboratory animals like guinea pigs by restricting tick feeding time. This approach might simulate natural tick bites and, if successful in humans, would mark a groundbreaking vaccine that targets the vector rather than the pathogen itself. However, as anti-tick vaccines are still evolving for human use in preventing tick-borne diseases, careful considerations regarding immunization strategies and antigen selection remain essential.

#### 7.2.4. Viral Vector-Based Vaccines

Viral vector vaccines use viruses as delivery systems to express target antigens within host cells, intending to trigger an immune response against them. These vaccines typically utilize viral vectors—like adenoviruses, vaccinia viruses, poxviruses, or flaviviruses—modified to transport and express genes encoding specific antigens. These vaccines provide several advantages, including the stimulation of a broad immune response and the expression of properly folded epitopes with optimal glycosylation, enhancing the immune response. Additionally, they can be administered orally [287]. Oral delivery holds potential for developing bait vaccines targeting wild animals like rodents or other small reservoir hosts involved in the transmission of tick-borne pathogens. Furthermore, they facilitate the delivery of multiple antigens, making them a compelling option for creating an anti-tick vaccine that is effective against multiple tick species. A.J. Ullmann et al. [187] explored the potential of adenoviral-vectored proteins in eliciting an immune response against *Ixodes scapularis* ticks and Lyme borreliosis. Their findings suggest that vaccinating with tick salivary proteins (SALP) via an adenoviral vector can effectively modulate a Th1 response in the host and partially control spirochete load in vaccinated mice following an infected tick challenge. The study underscores the promise of viral-vectored anti-tick vaccines as a promising avenue in research aimed at controlling tick populations and mitigating the transmission of tick-borne diseases.

## 8. Concluding Thoughts

Ticks and the diseases they transmit present a significant global challenge for both human and animal health. The primary measures employed to protect against tick bites currently involve wearing adequate clothing to cover exposed skin or using chemicals like permethrins, which can be harmful to various non-targeted species. While human vaccines against tick-borne pathogens, such as those for Lyme disease, are in development, an immense advantage of anti-tick vaccines is their potential to significantly reduce the population of several medically relevant ticks. More crucially, these vaccines could curtail the transmission of multiple diseases to humans and other larger hosts.

Targeting small animal reservoirs involved in the enzootic cycle of most tick-borne pathogens could prove to be an interesting strategy by which to disrupt the cycle at its early stages. This approach holds promise in preventing infections in nymphs or adult ticks, which are primarily responsible for transmitting diseases to humans. However, effectively combating this challenge requires a comprehensive strategy integrating diverse control methods. This includes managing cattle, the thoughtful use of acaricides, and the development of vaccines targeting multiple tick species. The pursuit of effective anti-tick vaccines is a complex journey marked by the intricate nature of tick–host–pathogen interactions.

A significant challenge lies in pinpointing and assessing suitable antigen candidates, relying on the intricate molecular mechanisms governing these interactions. In this context, proteins derived from tick salivary glands, midguts, eggs, and their secreted cement have emerged as promising targets for vaccine development. Several noteworthy candidates—such as SUB, MPs, ribosomal protein P0, serpins, and ferritin proteins like FER1, FER2, and aquaporins—have exhibited varying degrees of efficacy in curtailing tick infestations and limiting the transmission of associated pathogens.

While these antigens hold promise for anti-tick vaccine development, continuous research is essential in order to comprehensively grasp their mechanisms and refine their applications, including potential combinations. This optimization is vital to augment their effectiveness in controlling tick infestations and reducing the impact of tick-borne diseases on both humans and animals. Undoubtedly, advancements in genomics and proteomics over the past few decades have accelerated the discovery of new antigens and simplified protein manipulation, thereby streamlining vaccine testing procedures.

Various vaccine types have been extensively investigated in the quest for effective tick control strategies. DNA-based and mRNA-based vaccines have exhibited their potential by eliciting robust immune responses against tick antigens in controlled laboratory settings. Viral-vector-based vaccines, utilizing modified viruses to express specific tick antigens, have demonstrated the capability to induce broad immune reactions. Meanwhile, protein-based vaccines, such as the Bm86-based commercial vaccine, have shown promising results in reducing tick populations. However, challenges persist in achieving substantial reductions in tick populations through these individual approaches, as complete tick eradication remains an unrealistic goal due to matters of ecological balance.

The landscape of anti-tick vaccines presents a mosaic of challenges and potential solutions. Key milestones include optimizing antigen selection, addressing vaccine resistance in tick populations, and overcoming production and delivery barriers. Continued research and refinement are crucial to unlocking the full potential of anti-tick vaccines. These efforts hold the promise of not only managing tick populations but also significantly reducing the impact of tick-borne diseases on human and animal communities worldwide. Given the complexities of tick-borne disease transmission, a comprehensive approach is essential, one that integrates state-of-the-art research, innovative vaccine development, and robust implementation strategies to effectively tackle this global health challenge.

## Figures and Tables

**Figure 1 vaccines-12-00141-f001:**
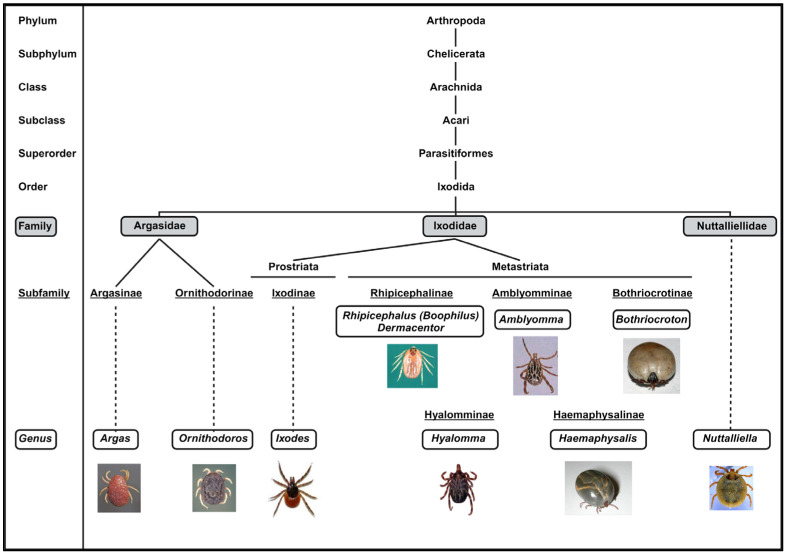
Diagram illustrating the taxonomy of ticks (NCBI taxonomy ID numbers: txid34601 (*Argas*), txid6937 (*Ornithodoros*), txid6944 (*Ixodes*), txid34630 (*Rhipicephalus*), txid34619 (*Dermacentor*), txid6942 (*Amblyomma*), txid189200 (*Bothriocroton*), txid34625 (*Hyalomma*), txid34622 (*Haemaphysalis*), and txid1029658 (*Nuttalliella*)).

**Figure 2 vaccines-12-00141-f002:**
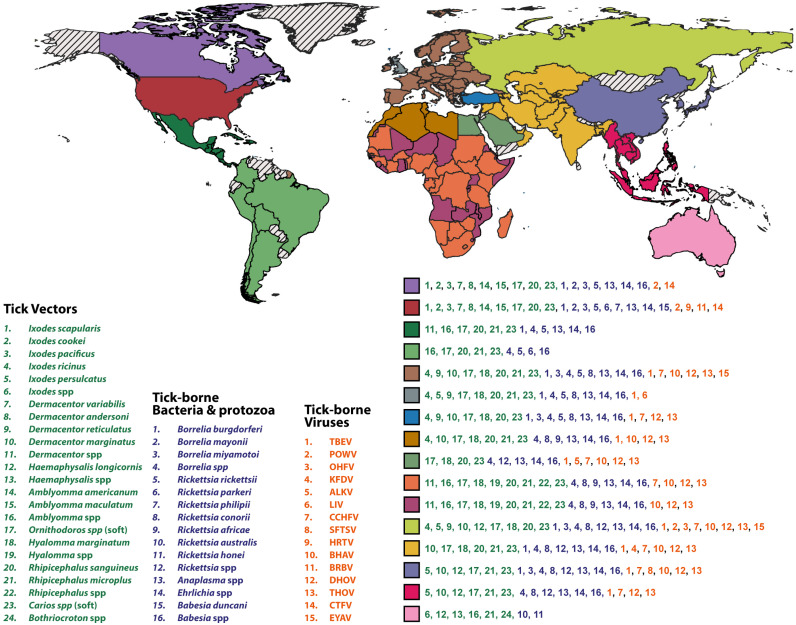
Geographical distribution of primary tick vectors and tick-borne pathogens. Green numbers represent the tick vectors, violet numbers denote the tick-borne bacteria and protozoa, and orange numbers indicate tick-borne viruses [25,26,27,28,29,30,31].

**Table 1 vaccines-12-00141-t001:** List of the different human tick-borne pathogens and their principal vector and distribution.

Associated Disease/Agent	Main Tick Vector (s)	Distribution by Country	Family/Genus
Tick-borne encephalitis (TBE)/TBE virus (TBEV) subtypes Siberian, Far East and European	*Ixodes ricinus*, *Ixodes persulcatus*	Europe, Asia, Middle East	*Flaviviridae*/*Orthoflavivirus*
Powassan encephalitis (POW)/POW virus (POWV)	*Ixodes scapularis*, *Ixodes cookei*, *Haemaphysalis longicornis*	Northeastern USA, Canada, Russian Far East	
Omsk hemorrhagic fever (OHF)/OHF virus (OHFV)	*Dermacentor reticulatus*, *Dermacentor* *marginatus*, *Ixodes persulcatus*	Western Siberia	
Kyasanur Forest Disease (KFD)/KFD virus (KFDV), Nanjianyin virus	*Haemaphysalis spinigera*	Indian subcontinent	
Alkhurma hemorrhagic Fever/Alkhurma virus (ALKV)	*Ornithodoros savignyi* (soft tick)	Middle East (Saudi Arabia, Egypt)	
Louping Ill/Louping Ill Virus (LIV)	*Ixodes ricinus*	Great Britain	
Crimean–Congo hemorrhagic fever (CCHF)/CCHF virus (CCHFV)	*Hyalomma marginatum* and >10 other tick spp.	Europe, Central Asia, India, Africa; expanding	*Nairoviridae/Orthonairovirus*
Severe fever with thrombocytopenia syndrome (SFTS)/SFTS virus (SFTSV) or Dabie bandavirus	*Haemaphysalis longicornis*, *Amblyomma testudinarium*, *Ixodes nipponensis* and *Rhipicephalus microplus*	China, Korea, Japan	*Phenuiviridae/Bandavirus*
Heartland virus infection/Heartland virus (HRTV)	*Amblyomma americanum*	Mid-western and southern USA	
Bhanja virus (BHAV) infection/BHAV	*Haemaphysalis* spp., *Dermacentor* spp., *Amblyomma* spp., *Rhipicephalus* spp. and *Hyalomma* spp.	Africa, Central Asia, southern Europe	
Bourbon virus (BRBV) disease/BRBV	*Amblyomma americanum*	USA	*Orthomyxoviridae/* *Thogotovirus*
Dhori virus (DHOV) infection/DHOV	*Hyalomma marginatum*	Africa, Asia, Middle East, Europe	
Thogoto virus (THOV) infection/THOV	*Haemaphysalis longicornis*, *Rhipicephalus* spp.		
Colorado tick fever (CTF)/CTF virus (CTFV)	*Dermacentor andersoni* and other tick spp.	Western USA and Canada	*Reoviridae/Coltivirus*
Eyach virus (EYAV) disease/EYAV	*Ixodes ricinus*, *Ixodes ventalloi*	Central Europe	
Lyme disease/*Borrelia burgdorferi* sensu stricto, *B. Mayonii*, *B. garinii*, *B. afzelii*, *B. bavariensis* and other *Borrelia* spp.	*Ixodes scapularis*, *I. pacificus*, *I. ricinus*, *I. persulcatus*	North America (*Borrelia burgdorferi* sensu stricto & *B. Mayonii* only), Europe, Asia	*Order: Spirochaetales* *Spirochaetaceae/Borrelia (borreliosis and relapsing fever)*
Relapsing fever borreliosis/ *B. miyamotoi*		North America, Europe, Asia	
Tick-borne relapsing fever (TBRF)/*Borrelia* spp.	*Ornithodoros* spp. and *Carios* spp. (soft ticks)	Worldwide (except Australia), mostly tropical and desert regions	
Southern tick-associated rash illness (STARI)/Unknown, *Borrelia lonestari* is a suggested candidate	*Amblyomma americanum*	Southern and eastern USA	
Rocky Mountain spotted fever (RMSF)/ *Rickettsia rickettsi*	*Dermacentor variabilis*, *D. andersoni*, *Rhipicephalus sanguineus*	America	*Order: Rickettsiales* *Rickettsiacae/Rickettsia*
Mediterranean spotted fever (MSF)/ *Rickettsia conorii*	*Rhipicephalus sanguineus*	Europe, Africa, Middle East, Asia	
*Siberian Tick Typhus/Rickettsia siberica*	*Dermacentor* spp.	China, Mongolia, Russia	
Queensland tick typhus/*Rickettsia australis*	*Ixodes holocyclus*, *I. tasmania*	Australia	
African tick bite fever (ATBF)/ *Rickettsia africae*	*Amblyomma* spp.	Coastal and sub-Saharan countries of Africa, West Indies	
Maculatum infection/*Rickettsia parkeri*	*Amblyomma maculatum*	North and South America	
Pacific Coast tick fever (PCTF)/ *Rickettsia philipii*	*Dermacentor occidentalis*	California, Pacific coast	
Flinders Island spotted fever, Thai tick typhus/*Rickettsia honei*	*Bothiocroton hydrosauri*	Australia, Thailand	
Human granulocytic anaplasmosis (HGA)/*Anaplasma phagocytophilum*	*I. scapularis*, *I. pacificus* (America), *I. ricinus* (Europe) and *I. persulcatus* (Asia)	Canada, Northeastern and central USA	*Order: Rickettsiales* *Anaplasmataceae/Anaplasma*
Human monocytic ehrlichiosis (HME)/ *Ehrlichia chaffeensis*	*Amblyomma americanum* (North America), other tick spp. including *Haemaphisalis* *longicornis* and *Rhipicephalus sanguineus* (Africa, Europe and Asia)	North America, Africa, Europe, Asia	*Order: Rickettsiales* *Anaplasmataceae/Ehrlichia*
Human ewingii ehrlichiosis (HEE)/ *E. ewingii*		North America, Europe, Asia	
Human ehrlichiosis/ *Ehrlichia muris eauclairensis*	*Ixodes scapularis*	Minnesota and Wisconsin, USA	
Neoehrlichiosis/*Neoehrlichia mikurensis*	*I. ricinus* and *I. persulcatus*	Europe, Asia	*Order: Rickettsiales* *Anaplasmataceae/* *Neoehrlichia*
Human babesiosis/*B. divergens*,	*I. ricinus*	Europe and Asia	*Babesiidae/Babesia*
Human babesiosis/*B. microti*	*I. scapularis*	North America	
Human babesiosis/*B. duncani*	*Dermacentor albipictus* and *I. scapularis*	North America (West coast)	
Human babesiosis/*B. venatorum*	*I. ricinus*, *I. persulcatus*	Europe, Asia	

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
