# Peer review of "Human Tick-Borne Diseases and Advances in Anti-Tick Vaccine Approaches: A Comprehensive Review"

_vaccines, 2024, doi:10.3390/vaccines12020141_

Round 1

Reviewer 1 Report

Comments and Suggestions for Authors

Unfortunately, I did not like the idea of the manuscript of Nepveu-Traversy et al. “Tick-Borne Diseases and Advances in Anti-Tick Vaccine Approaches: A Comprehensive Review” to combine two global and poorly connected parts in one review article: i) a description of ticks and all tick-borne human infections (viral, bacterial and protozoan) and ii) a description of anti-tick vaccines. The topic “Anti-tick vaccine” should be written as a separate review article, and not as a small part of it (only 8 pages). Moreover, the topic of “tick-borne infections” is too global even for one review article. The way this part of the article is written, it reports brief and inconsistent data for different agents. In addition, the description of ticks and tick-borne agents contains many errors.

In my opinion, the review article should be fundamentally rewritten. The topic “anti-tick vaccine” should be written in more detail. Thus, the results of the practical use of vaccines in the field and the description of candidate antigens for vaccines were very briefly described. Not all candidate antigens (eg, vitelin) were discussed. To facilitate understanding of this article, a table summarizing the antigen candidates should be included (e.g., similar to the table in Abbas et al. Int J Mol Sci. 2023 4;24(5):4969. doi: 10.3390/ijms24054969).

As for the first part of the manuscript, a description of ticks and tick-borne pathogens should be given only as a brief introduction to the topic “Anti-tick vaccines”. There is no need to describe all agents, but only the most significant ones. Moreover, since vaccines can be used to vaccinate farm animals, but not humans, it is also necessary (and even primarily) to describe the causative agents of tick-borne infections in farm and domestic animals.

Specific comments

1. The title of manuscript should be changed to focus on tick-borne vaccines.

2. Line 63. Notable genera in the Ixodidae family include not only Ixodes, Rhipicephalus, and Amblyomma, but also Dermacentor, Haemaphysalis and Hyalomma.

3. The Figure 1 is not quite correct; Rhipicephalus, Amblyomma, Dermacentor, and Bothriocroton are not located at the level of genus, whereas Haemaphysalinae and Hyalomminae are not located at the level of subfamily.

3. Lines 71, 73 and throughout the text. Ixodid and argasid are not taxonomic terms and must begin with a lowercase letter

4. line 71. Not all ixodid ticks are three-host ticks

5. Line 97. Not all agents can be transmitted transovarially.

6. Lines 103 and throughout the text. The current name for Boophilus is Rhipicephalus.

7. Table 1. It is unclear what criteria define prevalence as rare or common. Does this mean the prevalence of the pathogen in ticks or infection in humans? The prevalence of the same pathogen may be rare in one region and common in another. If the authors are referring to human infection, then the prevalence of relapsing fever borreliosis and Siberian Tick Typhus in Russia is high, while the prevalence of HGA, HME and HEE in Europe is very low. The Table contains some errors. Rocky Mountain spotted fever was not common in Western Europe. Ixodes persulcatus is the main vector of Neoehrlichia mikurensis in Asia.

8. Lines 161-168. The association of different TBEV subtypes with clinical symptoms and vectors is not strict

9. Lines 344-346. It’s strange to give trivial names of ticks (sheep tick, taiga tick, black-legged tick) only on page 10 of the manuscript, and not at the beginning.

10. Line 347. Borrelia bavariensis is also a highly pathogenic and widespread causative agent of Lyme disease in Eurasia.

11. Line 393. The vaccine for Lyme borreliosis was not safe; it caused complications in the form of arthritis and autoimmune rheumatoid arthritis.

12. Line 402-406. In this section, the authors should add the reference on Platonov et al. (2011), who first proved the pathogenic properties of Borrelia miyamotoi.

Comments on the Quality of English Language

The quality of English is very good. I have only noted minor errors (e.g. Antigens Candidates instead of Antigen Candidates)

Author Response

Response to Reviewer #1 Comments

We are grateful for your thorough review and suggestions regarding the structure of our manuscript. We appreciate your perspective on the integration of the topics of tick-borne diseases and anti-tick vaccines. All the comments we received on this study have been taken into account toward improving the quality of the article, and we present our reply to each of them separately. We are confident that these additions significantly improve the manuscript, making it a more thorough and valuable resource for readers interested in the field of anti-tick vaccines. We appreciate your guidance in making these important improvements and hope that the revised manuscript now meets the high standards of your journal.

Once again, we are grateful for your constructive critique and the opportunity to enhance our work based on your insights.

Major comment 1:

Unfortunately, I did not like the idea of the manuscript of Nepveu-Traversy et al. “Tick-Borne Diseases and Advances in Anti-Tick Vaccine Approaches: A Comprehensive Review” to combine two global and poorly connected parts in one review article: i) a description of ticks and all tick-borne human infections (viral, bacterial and protozoan) and ii) a description of anti-tick vaccines. The topic “Anti-tick vaccine” should be written as a separate review article, and not as a small part of it (only 8 pages).

Answer: We thank the reviewer for pointing out the apparent lack of connection between the initial part of the review and the section on tick-borne vaccines. Based on the reviewer’s comments, we have improved the structure of the manuscript to better connect the different topics reviewed. The revised manuscript is now divided into four major topics, including an overview of 1) tick species, 2) human pathogens they transmit, 3) existing countermeasures (Section 6 was added to the revised manuscript) and finally 4) Vaccines. Our rationale for combining these topics is grounded in the following reasons: Interconnectedness of Topics: We believe that discussing tick-borne diseases provides essential context for understanding the development and significance of anti-tick vaccines. Comprehending the diversity and impact of these diseases is crucial to appreciating the need for, and mechanisms of, current and emerging vaccines.

Holistic Approach: Our intention is to provide a comprehensive overview that bridges the gap between disease understanding and preventative measures. By presenting these topics together, we aim to offer readers, especially those new to the field, a more complete understanding of the challenges and solutions in tick-borne disease management.

We are confident that these additions significantly improve the manuscript, making it a more thorough and valuable resource for readers interested in the field of anti-tick vaccines. We appreciate your guidance in making these important improvements and hope that the revised manuscript now meets the high standards of your journal.

Once again, we are grateful for your constructive critique and the opportunity to enhance our work based on your insights.

Major comment 2:  

Moreover, the topic of “tick-borne infections” is too global even for one review article. The way this part of the article is written, it reports brief and inconsistent data for different agents. In addition, the description of ticks and tick-borne agents contains many errors.

Answer: We concur with Reviewer 's observation that a single review cannot encompass every tick-borne pathogen. Our review specifically focuses on human tick-borne pathogens from each microorganism (viruses, bacteria, and parasites) that have the potential to spread to extensive geographical areas and pose a significant future health burden. We have highlighted this point in the revised manuscript (lines 130-133).

Therefore, we refined our manuscript to improve coherence, address any inconsistencies, and correct factual errors, particularly in the descriptions of ticks and tick-borne agents.

Major comment 3:  

In my opinion, the review article should be fundamentally rewritten. The topic “anti-tick vaccine” should be written in more detail. Thus, the results of the practical use of vaccines in the field and the description of candidate antigens for vaccines were very briefly described. Not all candidate antigens (eg, vitelin) were discussed. To facilitate understanding of this article, a table summarizing the antigen candidates should be included (e.g., similar to the table in Abbas et al. Int J Mol Sci. 2023 4;24(5):4969. doi: 10.3390/ijms24054969).

Answer: We also thank you for valuable feedback regarding the inclusion of additional antigens in our manuscript. We agree with your observation that our initial submission did not comprehensively cover all relevant antigens in the context of anti-tick vaccines.

In response to your suggestion, we have revisited the relevant sections of our manuscript and have now included additional antigens (cystatins, Isac, tHRF, TSLPI, TIX-5, UBQ, ATAQ, Ba86, Bm95, Haa86, 5’-nuleotidase, BYC, GP80, Vitellin, VTDCE, ABC transporter, selenoproteins, calreticulin and cathepsin L-like cysteine proteinase) that play a significant role in the development of anti-tick vaccines. These additions not only enrich the content but also ensure a more comprehensive coverage of the topic.

Furthermore, in line with your recommendation, we have added a new table summarizing the candidate antigens for anti-tick vaccines (Table 2 in the revised manuscript). The format is inspired by Abbas et al. (Int J Mol Sci. 2023, 24(5):4969). Notably, we have included information on the efficacy of protection against both ticks and pathogen transmission, when available. This addition is expected to significantly improve the readability and comprehension of our review. The table serves as a quick reference, offering a consolidated view of various antigens, their characteristics, and their relevance in vaccine development.

Major comment 4:

As for the first part of the manuscript, a description of ticks and tick-borne pathogens should be given only as a brief introduction to the topic “Anti-tick vaccines”. There is no need to describe all agents, but only the most significant ones.

Answer: As mentioned in our reply to comment 2 and as suggested by this reviewer, our review specifically focuses on human tick-borne pathogens from each microorganism (viruses, bacteria, and parasites) that have the potential to spread to extensive geographical areas and pose a significant future health burden. We have, however, ensured that this section is succinct and directly relevant to the broader context of anti-tick vaccines. We hope that this approach will be seen as a valuable contribution to the understanding of the full spectrum of tick-borne diseases and their implications for vaccine development.

Major comment 5:

Moreover, since vaccines can be used to vaccinate farm animals, but not humans, it is also necessary (and even primarily) to describe the causative agents of tick-borne infections in farm and domestic animals.

Answer: Thank you for your suggestion. While recognizing the significance of tick-borne diseases in animals, our review specifically targets pathogens of public health concern, focusing on their impact on human populations and related vaccine development. This specific angle was chosen to address the urgent need for public health solutions. However, we have briefly acknowledged the relevance of these diseases in farm and domestic animals, underscoring the potential for vaccine applications in veterinary medicine, while maintaining our primary focus on human health in the Anti-Tick Countermeasures section (lines 647-653).

We understand that the structure of a review can significantly impacts its clarity and effectiveness.

We hope this response clarifies our decision to maintain the combined structure of the manuscript. We believe that with the proposed enhancements, the article will serve as a valuable resource for readers interested in both the challenges posed by human tick-borne diseases and the evolving landscape of anti-tick vaccines.

Minor Comment 1: The title should be more focused on tick-borne vaccines.

Response:  While we believe that the current title accurately captures the essence of the topic, we have added 'human' to emphasize that this comprehensive review exclusively addresses pathogens and associated tick-borne diseases affecting humans. This adjustment aims to prevent any potential confusion. The revised title ensures a balanced and informative overview, contributing valuable insights to the field.

Minor comment 2: Line 63. Notable genera in the Ixodidae family include not only Ixodes, Rhipicephalus, and Amblyomma, but also Dermacentor, Haemaphysalis and Hyalomma.

Response: We appreciate your attention to detail in pointing out this inaccuracy. The mentioned line has been removed, as it was neither accurate nor relevant to the text.

Minor comment 3: The Figure 1 is not quite correct; Rhipicephalus, Amblyomma, Dermacentor, and Bothriocroton are not located at the level of genus, whereas Haemaphysalinae and Hyalomminae are not located at the level of subfamily.

Response: In support of the taxonomy depicted in Figure 1, we have included all NCBI taxonomy identification numbers in the title. We acknowledge that variations may exist with other sources of information, but we deem this data to be both relevant and up-to-date. Any suggestions regarding official authorities or references for taxonomy, if available, would be greatly appreciated.

Minor comment 4: Issues with the use of ixodid and argasid terms throughout the text.

Response: Thank you for pointing out the incorrect usage of these terms, we have revised these sections to ensure that terms like 'ixodid' and 'argasid' start with lowercase letters.

Minor comment 5: line 71. Not all ixodid ticks are three-host ticks

Response: Indeed, we recognized that this section was potentially confusing, and as a result, we have made adjustments (lines 73-77) to ensure clarity regarding this information.

Minor comment 6: Line 97. Not all agents can be transmitted transovarially.

Response: We acknowledge the oversight in our description of pathogen transmission methods. The revised manuscript has been amended to explicitly state that not all pathogens are transmitted in this manner (lines 103-104).

Minor comment 7:  Lines 103 and throughout the text. The current name for Boophilus is Rhipicephalus..

Response: Thank you for highlighting the need to update our nomenclature. We have revised the manuscript to consistently use the current name 'Rhipicephalus' instead of 'Boophilus' throughout the entire document.

Minor comment 8: Issues with Table 1 regarding the criteria for defining prevalence and errors in the prevalence or distribution of certain diseases.

Response: We appreciate your critical evaluation of Table 1. In response, we have revised the table, removing the section on defining the prevalence of pathogens in ticks or human infection due to its confusing nature. Furthermore, we have corrected geographical inaccuracies and updated tick species to align with the most current data.

Minor comment 9: Association of TBEV subtypes with clinical symptoms and vectors is not strict.

Response: Thank you for pointing out the need for clarity regarding the association of TBEV subtypes with clinical symptoms and vectors. We acknowledge that our initial representation might have oversimplified this relationship. In the revised manuscript, we have modified to better reflect the complexity and variability in the association of TBEV subtypes with clinical symptoms and vectors (lines 173-178). We now provide a more nuanced discussion that highlights the current understanding of these associations, as well as areas where uncertainties still exist.

Minor comment 10: Trivial names of ticks introduced late in the manuscript.

Response: We appreciate your observation regarding the introduction of the trivial names of ticks. We have now revised the manuscript and removed these trivial names, ensuring a more logical flow and immediate clarity for our readers.

Minor comment 11: Borrelia bavariensis as a pathogen of Lyme disease in Eurasia.

Response: Thank you for highlighting the omission of Borrelia bavariensis as a significant causative agent of Lyme disease in Eurasia. This was an inadvertent oversight in our review. We have included this pathogen in Table 1 and amended line 360 to include Borrelia bavariensis, acknowledging its role as a highly pathogenic and widespread agent. This addition ensures a more accurate and comprehensive coverage of Lyme disease pathogens."

Minor comment 12: Safety concerns of the vaccine for Lyme borreliosis.

Response: We are grateful for your attention to the safety profile of the Lyme borreliosis vaccine. Your comment has prompted us to re-examine and update our discussion on this topic. In the revised manuscript, we now added a sentence at lines 407-410: “A recombinant OspA-based vaccine was introduced in the United States from 1998 to 2002. However, concerns arose regarding its safety, particularly its potential relationship to autoimmune arthritis, which led to its eventual withdrawal from the market”.

Minor comment 13: Reference to Platonov et al. (2011) on Borrelia miyamotoi.

Response: Thank you for suggesting the addition of the reference to Platonov et al. (2011), which discusses the pathogenic properties of Borrelia miyamotoi. This reference is indeed pertinent to our discussion and should have been included in our initial submission. We have now incorporated this reference in the relevant section (line 421).

Comments on the quality of English language: Antigens Candidates instead of Antigen Candidates

Response: Thank you for bringing attention to those typos; they have been corrected in the revised manuscript.

Reviewer 2 Report

Comments and Suggestions for Authors

This review article entitled ‘’Tick-borne diseases and advenges in anti-tick vaccines approaches: a comprehensive review’’ presents a comprehensive review regarding the status of tick-borne diseases and anti-tick vaccines. It provides a brief overview of current vaccine approaches for the tick control, addressing their significance in combating tick-borne diseases of public health concern. The main objectives are to provide a brief epidemiology of diseases and a thorough understanding of tick biology, traditional tick control methods, the development and mechanisms of anti-tick vaccines. Importantly, this review also highlights important informations and major gaps in scientific knowledge on their efficacy in field applications, associated challenges, and future prospects. It is intended to provide a reference for all those interested in the importance of developing and implementing anti-tick vaccines in order to reduce the use of acaricides and, ultimately, reduce acaricide resistance. As a result, it is a comprehensive, well-organized, and well-written review article. Appropriate referencing and a thorough review of the current status of major tick-borne diseases and anti-tick vaccine development.

A few minor comments

-Line 63; it is stated that notable genera in the Ixodidae family include Ixodes, Rhipicephalus, and Amblyomma. It is okey, this determination can be made, but saying that these are more important may vary in different situations. For example, the genus Hyalomma is very important for both veterinary and public health. Because the species in this genus act as vectors for T. annulata, which is an important disease of animals, and the Crimean-Congo hemorrhagic fever virus, which is very important for humans. When making such judgments, authors must consider the subject in its entirety. Ultimately, the information provided in this review should attract the attention of researchers working in different scientific disciplines.

-Line 83; there are many species of ixodid ticks that are found in larval and nymphal stages on small mammals as well as ground-feeding birds (Hyalomma spp., Amblyomma spp.). In my opinion, it would be better if ground-feeding birds were added to the end of this sentence.

-Lines 102-104; ……. feed and mature on the same host, especially on cattle.

- Table 1; Babesia venatorum is an emergeing protozoon human disease in some poart of Europe. It would be better if it is added to the relevant place in Table 1.

Author Response

Response to reviewer #2

We thank Reviewer #2 for their thorough and constructive feedback on our manuscript titled "Tick-borne diseases and advances in anti-tick vaccines approaches: a comprehensive review." Your insights have significantly contributed to the enhancement of our paper, and we have carefully considered and incorporated your suggestions. We believe that these revisions have substantially improved the manuscript and hope that they adequately address your concerns. We are grateful for the opportunity to refine our work based on your valuable insights.

Below, we provide a detailed response to each of your comments, along with explanations of how we have revised the manuscript accordingly:

Minor comment 1: Line 63; it is stated that notable genera in the Ixodidae family include Ixodes, Rhipicephalus, and Amblyomma. It needs some explanation.

Response: We appreciate your attention to detail in pointing out this inaccuracy or incomplete information. The mentioned line has been removed, as it was neither accurate nor relevant to the text.

Minor comment 2: Line 83: The reviewer recommends adding ground-feeding birds to the list of hosts for ixodid ticks in their larval and nymphal stages.

Response: Thank you for your suggestion. We have incorporated the provided information into the text (lines 87-89), enhancing the accuracy and reflecting the reality more effectively.

Minor comment 3: Lines 102-104, the reviewer suggests to add: “especially on cattle” at the end of the sentence.

Response: We have made the necessary correction as advised, specifically in line 110.

Minor comment 4: Table 1; Babesia venatorum is an emerging protozoan human disease in some part of Europe. It would be better if it is added to the relevant place in Table 1.

Response: This emerging pathogen in Europe and Asia has been added to Table 1, as suggested, and incorporated into the manuscript (lines 553-555). This addition better reflects the significance of this tick-borne pathogen.

Round 2

Reviewer 1 Report

Comments and Suggestions for Authors

The authors have significantly improved their review article. In the revised version, the section “Advances in Anti-Tick Vaccines” is written in sufficient detail and contains an informative table. However, the main goal of anti-tick vaccines in the near future is to minimize the number of ticks feeding on cattle. Protecting people from tick bites and tick-borne pathogens using anti-tick vaccines is unlikely to be realized in the coming years. Thus, in my opinion, authors should include at least brief information about pathogens of farm animals.

Specific comments

1. In Europe, Borrelia venatorum is transmitted by Ixodes ricinus.

2. Line 355. Haemaphysalis, Dermacentor, Hyalomma and Rhipicephalus transmit TBEV in rare cases.

3. Lines 654-656 and throughout the text. The species name should not be capitalized (as was in text, B. Miyamotoi, B. Burgdorferi sensu lato, I. Pacificus).

4. Line 684. Add, please, “and neoehrlichiosis”.

5. Lines 958-963 “While tick-borne diseases undoubtedly impact both farm and domestic animals, vaccination holds the potential to safeguard not only hese animals but also humans. By targeting reservoir hosts, including wild and farm animals, with veterinary vaccines, we can not only benefit animal health and agriculture but also disrupt the transmission of various tick-borne diseases to humans. This approach has the dual advantage of protecting both animals and humans from the harmful effects of these diseases.” - This is a wonderful, but unrealistic project.

6. Line 1192 “Blocking this interaction could potentially reduce the transmission of the pathogen to vaccinated hosts”;

Line 1220 “SUB, found across various tick species, has shown promise in vaccination efforts, conferring protective effects against multiple ticks and pathogen transmission”;

Line 1228 “ …reduce engorgement and B. borrelia transmission in animals immunized with the recombinant proteins’ – Indicate, please, which animals were vaccinated in the above cases.

Author Response

Response to Reviewer #1 Comments (Round 2)

We thank the reviewer for his continued engagement and the additional constructive comments regarding our manuscript titled " Human Tick-Borne Diseases and Advances in Anti-Tick Vaccine Approaches: A Comprehensive Review." His insights have been instrumental in refining our work, and we are thankful for the opportunity to further enhance its quality. We have carefully considered the recent queries raised and have addressed each in detail. Below, I provide a point-by-point response to ensure that we have fully met your concerns and the standards of Vaccines journal. We believe that these clarifications and additional details will contribute to a more comprehensive understanding of our research on human tick-borne diseases and anti-tick vaccines. Our team is committed to presenting work that is both rigorous and informative, and we hope that these adjustments align with your expectations and the journal's standards.

Once again, I extend my gratitude for your meticulous review and valuable feedback. We are eager to see our manuscript contribute meaningfully to the field and are optimistic that it now aligns closely with the journal's aims and scope.

Major comment 1:  The authors have significantly improved their review article. In the revised version, the section “Advances in Anti-Tick Vaccines” is written in sufficient detail and contains an informative table. However, the main goal of anti-tick vaccines in the near future is to minimize the number of ticks feeding on cattle. Protecting people from tick bites and tick-borne pathogens using anti-tick vaccines is unlikely to be realized in the coming years. Thus, in my opinion, authors should include at least brief information about pathogens of farm animals.

Answer: We thank reviewer for his constructive feedback and for recognizing the improvements made in the revised version of our review article. We are pleased to hear that the details provided in the section “Advances in Anti-Tick Vaccines” have met your expectations.

We acknowledge your insightful suggestion regarding the inclusion of information about pathogens affecting farm animals. We now added the short paragraph between the Lines: 141-157 in the section 3.

This paragraph will briefly outline the most prevalent tick-borne pathogens affecting cattle and other livestock, their impact on animal health and farm productivity, and current strategies used to mitigate the risk associated with these pathogens. We believe that this addition will provide a more comprehensive understanding of the importance of anti-tick vaccines in the context of livestock health.

Minor comment 1:  In Europe, Borrelia venatorum is transmitted by Ixodes ricinus.

Answer: Indeed, the information in the table was incomplete, and we have since modified it to include I. ricinus.

Minor comment 2:  Line 355. HaemaphysalisDermacentorHyalomma and Rhipicephalus transmit TBEV in rare cases.

Answer: Thank you for bringing attention to the imprecise information. To enhance clarity, we have revised the statement to specify that these ticks are recognized as vectors in Asia, as they have the capability to carry and transmit the virus (lines 195-196).

Minor comment 3:  Lines 654-656 and throughout the text. The species name should not be capitalized (as was in text, B. MiyamotoiB. Burgdorferi sensu lato, I. Pacificus).

Answer: All text was monitored to correct those typos; all species names are now spelled without a capital letter throughout the text.

Minor comment 4:  Line 684. Add, please, “and neoehrlichiosis”.

Answer: This information has been added as suggested, and it is definitely more accurate now (line 451).

Minor comment 5:   Lines 958-963 “While tick-borne diseases undoubtedly impact both farm and domestic animals, vaccination holds the potential to safeguard not only these animals but also humans. By targeting reservoir hosts, including wild and farm animals, with veterinary vaccines, we can not only benefit animal health and agriculture but also disrupt the transmission of various tick-borne diseases to humans. This approach has the dual advantage of protecting both animals and humans from the harmful effects of these diseases.” - This is a wonderful, but unrealistic project.

Answer: The original statement was potentially misleading, as pointed out by the reviewer. Instead, we have modified certain words to mitigate the perception of veterinary vaccines directly protecting human from pathogen transmission. The revised statement now reads as follows: “While tick-borne diseases undoubtedly impact both farm and domestic animals, vaccination holds the potential to safeguard not only these animals but also to have a significant impact on humans. By targeting reservoir hosts, including wild and farm animals, with veterinary vaccines, we can benefit not only animal health and agriculture but also significantly decrease the risk of transmission of various tick-borne diseases to humans”. (lines 668-674)

Moreover, we recognize several challenges in implementing a One Health approach that focuses on vaccinating animals against tick infestations to reduce the acquisition of tick-borne diseases. However, it's important to note the success of large-scale bait vaccination programs in Europe and North America, where immunizing wild foxes against rabies has significantly decreased rabies cases in nearby populations. This demonstrates the practicality of a similar strategy. The effectiveness of these rabies vaccination programs shows that, although feasible, such approaches require ongoing immunization of the target animal species.

Recent developments in vaccine technology increase the feasibility of implementing such projects on a large scale. We believe that, despite challenges, this approach holds significant potential for controlling tick-borne diseases in both animals and humans.

Minor comment 6:  Line 1192 “Blocking this interaction could potentially reduce the transmission of the pathogen to vaccinated hosts”;

Answer: The specification of the vaccinated animal, namely mice, has been added as suggested (line 788).

Minor comment 7:  Line 1220 “SUB, found across various tick species, has shown promise in vaccination efforts, conferring protective effects against multiple ticks and pathogen transmission”;

Answer: The specification of the vaccinated animal, namely cattle, has been added as suggested (line 811-812).

Minor comment 8:  Line 1228 “ …reduce engorgement and B. borrelia transmission in animals immunized with the recombinant proteins’ – Indicate, please, which animals were vaccinated in the above cases.

Answer: The specification of the vaccinated animals, namely cattle and rabbit, has been added as suggested (line 819).